

**The impact of coral reef ecosystems and upwelling events on the marine carbon**
**dynamics of Southern Taiwan**
Pei-Jie Meng[1-2*], Chia-Ming Chang[1], Hung-Yen Hsieh[1-2], Anderson B. Mayfield[2-3],
Chung-Chi Chen[4,1*]
[1]Graduate Institute of Marine Biology, National Dong Hwa University, Checheng,
Pingtung 944, Taiwan
[2]National Museum of Marine Biology and Aquarium, Checheng, Pingtung 944,
Taiwan
[3]Coral Reef Diagnostics, Miami, FL 33129, USA
[4]Department of Life Science, National Taiwan Normal University
88, Sec. 4, Ting-Chou Road
Taipei 11677, Taiwan
Running title: $p$CO$_2$ in a coral reef ecosystem
* Corresponding authors: pjmeng@nmmba.gov.tw; ccchen@ntnu.edu.tw
Email: ccchen@ntnu.edu.tw
Phone: 886-2-2930-2275
Fax #: 886-2-2931-2904



31              **ABSTRACT**

The ocean is the largest carbon reservoir and plays a crucial role in regulating
atmospheric $CO_2$ levels, especially in the face of climate change. In coral reef
ecosystems, the complexity and importance of the carbonate system must be better
appreciated as atmospheric $CO_2$ concentrations continue to rise. This study measured
$pCO_2$ over space and time in Nanwan Bay, a coral reef ecosystem in Southern
Taiwan, to identify factors that influence its variation. The results showed that mean
$pCO_2$ values varied seasonally, with values of 394, 406, 399, and 367 μatm in spring,
summer, fall, and winter, respectively. These seasonal mean differences ($\Delta pCO_2$)
relative to atmospheric $pCO_2$ (397, 392, 392, & 396, respectively) were -3, 14, 7, and
-29 μatm, respectively. These findings suggest that the Nanwan Bay coral reef
ecosystem acts as a sink for atmospheric $CO_2$ during the spring and winter, with an
average sea-air gas flux of -1 gC m$^{-2}$ year$^{-1}$ and a net annual uptake of -29 tC. The
carbonate system parameters of the surface water in this high-biodiversity sub-tropical
marine ecosystem were influenced not only by seasonal temperature variation but also
by vertical mixing, intermittent upwelling, and biological effects.

**Keywords**: carbon sink, carbon source, coral reef, $pCO_2$, total alkalinity, upwelling



## 1. Introduction


The concentration of atmospheric carbon dioxide ($CO_2$) varies significantly

based on region and season. High-latitude temperate regions and coastal seas act as
sinks for atmospheric $CO_2$, while subtropical and tropical coastal seas, estuaries, and
coral reefs are generally sources (Borges et al., 2005; Cai et al., 2003; Frankignoulle
et al., 1998; Frankignoulle et al., 1996; Gattuso et al., 1997; Gattuso et al., 1993; Ito et
al., 2005; Ohde and Van Woesik, 1999; Wang and Cai, 2004; Yan et al., 2011; Bates et
al., 2001). The hydrological characteristics of coastal waters can vary significantly,
leading to differences in surface water $p$CO$_2$ even within the same continental shelf.
Furthermore, upwelling areas (such as California & Oman) are sinks for $CO_2$, while
the coasts of Galicia and Oregon are sources (Borges and Frankignoulle, 2002;
Friederich et al., 2002; Goyet et al., 1998; Hales et al., 2005). Borges (2005) notes
that when estuaries are included in the gas exchange process, coastal seas worldwide
are sources of $CO_2$, becoming sinks when estuaries are excluded.

Various factors, such as temperature, tides, currents, river discharge, upwelling,

vertical mixing, and biological metabolism, can influence $CO_2$ levels in coastal areas
(e.g., Dai et al., 2009). These factors can interact and help explain why seasonal
variation in $CO_2$ levels can differ greatly across regions. For instance, measurements
taken at the Bermuda Atlantic Time-series station in the northwest Atlantic from 1996



to 1998 showed that $CO_2$ levels were lowest in winter and highest in summer
(Takahashi et al., 2002). Similarly, data collected from the Kyodo Western North
Pacific Ocean Time-series station between 1998 and 2000 indicated that $CO_2$ levels
were lower in summer compared to winter (Takahashi et al., 2002).

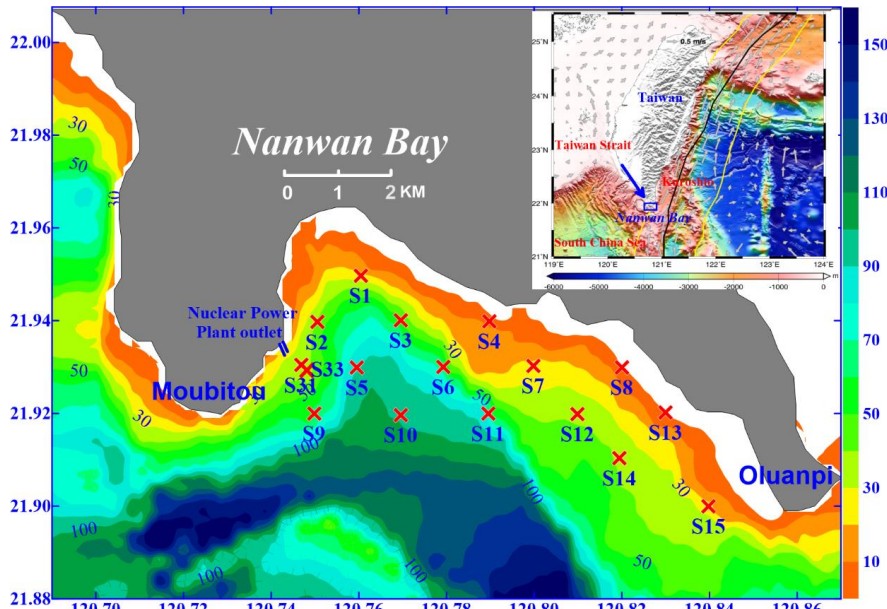


**Fig. 1** Map of sampling stations in Nanwan Bay, Taiwan. The color bar indicates the

bottom depth (m). The blue arrow in the upper right inset shows the sampling

region.

Coral reefs, despite occupying only 0.2% of the ocean, are home to a significant

portion of marine biodiversity, housing one-third of all marine species (Sheppard et
al., 2017; Reaka-Kudla, 1997). These productive ecosystems boast efficient and stable
carbonate sedimentation rates, contributing to 23-26% of global annual $CaCO_3$
sedimentation (Suzuki and Kawahata, 2004). However, coral reefs are highly



vulnerable to the effects of climate change and anthropogenic pollution, which can
significantly impact the organisms responsible for building the reef structure
(Bellwood et al., 2004; Hughes et al., 2017; Chen, 2021). Due to the structural
complexity and high biodiversity of coral reefs, their carbon dynamics may differ
substantially from those of the open ocean. Currently, it is unclear whether coral reef
ecosystems act as net carbon sources (Ware et al., 1992; Gattuso et al., 1993; Gattuso
et al., 1999; Fagan and Mackenzie, 2007; Lønborg et al., 2019; Yan et al., 2018;
Watanabe and Nakamura, 2019; Frankignoulle et al., 1998) or sinks (Kayanne et al.,
1995; Mayer et al., 2018; Suzuki, 1998; Suzuki and Kawahata, 2004). Additionally,
since environmental changes can result in physiological changes in resident organisms
(Fabry et al., 2008), it is challenging to predict how seawater carbon levels will
change in response to oceanographic anomalies.

Semi-enclosed Nanwan Bay is situated at the southernmost point of Taiwan,

(Fig. 1). It is flanked by the Pacific Ocean to the east and the Taiwan Strait to the
west, and faces the Luzon Strait to the south and the South China Sea (SCS) to the
southwest (Lee, 1999). The bay stretches between Cape Moubitou and Cape Oluanpi
and boasts a coastline primarily consisting of fringing coral reefs (Yang and Dai,
1980). Nanwan Bay is among the most diverse marine regions in Taiwan, which led to
its inclusion within Kenting National Park (Meng et al., 2008). The complex seabed in





101 Nanwan Bay comprises various habitats and is the first point of contact for the warm

102 and highly saline Kuroshio Current. The water is oligotrophic, and temperatures

103 typically range from 21 to 30°C, but with periodic upwelling events occurring (Chen

104 et al., 2005). The bay hosts over 1,200 fish species and more than 200 species of reef-

105 building corals, making it a significant research focus area for both the reefs and the

106 anthropogenic stressor regime (Meng et al., 2007). Studies have shown that high

107 levels of nutrients and suspended solids may have contributed to the decline in coral

108 cover between 2001 and 2022 (Meng et al., 2008; Chen et al., 2022).

109  In previous studies of Nanwan Bay, periodic upwelling caused mixing of upper

110 and lower seawater layers, which contributed significantly to the transfer of nutrients

111 from the depths to shallower areas (Chen et al., 2005). In some upwelling areas, $CO_2$

112 may be released into the atmosphere, while in others, $CO_2$ enters the ocean from the

113 atmosphere. Basic productivity in marine ecosystems has a potential impact on carbon

114 cycling (Dugdale and Wilkerson, 1989; Murray et al., 1995), and upwelling brings

115 nutrients into the photic zone, thereby stimulating the proliferation of phytoplankton

116 and enhancing basic productivity. As a result, there is also an increased demand for

117 carbon for $CO_2$ fixation at these times (Chen et al., 2004b). The average ratio of

118 primary production to community respiration (P/R ratio) is often used to determine

119 whether a marine system is a source or sink of $CO_2$ in the atmosphere (where P/R<1



indicates a source, & P/R>1 indicates a sink; Smith and Hollibaugh, 1993; Robinson
et al., 2002). We sought herein to determine whether Nanwan Bay is a net carbon
source or sink by characterizing the marine carbonate system over time and under a
range of biogeochemical processes.

**2.  Methods**

**2.1 Sampling and analysis** The study was conducted across four seasons: spring

(31 March 2011), summer (5 July 2011), autumn (20 October 2011), and winter (22
January 2013), in the area between Nanwan Bay's two capes, Cape Moubitou and
Cape Oluanpi. A total of 17 seawater sampling stations were established, including
one near the outlet of a nuclear power plant (Fig. 1). Temperature and salinity data,
both with accuracy of 99.9%, were collected using an Idronaut Ocean Seven 304 CTD
calibrated against an International Association for the Physical Sciences of the Ocean
seawater standard. Water samples were collected using Niskin bottles with Teflon-
coated inner walls. Seawater at each station was taken at two to five depths at
intervals of 3 to 25 m in areas shallower than 50 m; extra samples at 65, 80 and/or 100
m were taken for stations with depths of 65-100 m. Water samples were immediately
analyzed for dissolved oxygen (DO) content using YSI 52 and YSI 5905 BOD
electrodes (accuracy=99.9%). Other water samples were divided into different sample
bottles for additional analyses. One 300-mL amber bottle was pre-inoculated with 0.2



mL of mercuric chloride to suppress biological activity that could affect total
alkalinity (TA) and other carbonate system parameters.

Seawater pH and total TA were measured using an automated titration system

consisting of a Mettler-Toledo DL53 with a DG-111 electrode. Prior to measurement,
the electrode was calibrated using Merck standard buffer solution (NIST) at 25°C.
The calibration ranges for pH 4, 7, and 10 were set to fall within the range of 176±30
mV, 0±30 mV, and -176±30 mV, respectively (calibration slope of -56 to -59).
Measured values were expressed on the NBS scale ($pH_{NBS}$). The electrode was also
calibrated using Tris-artificial seawater buffered at pH 8.083 and AMP artificial
seawater buffered at pH 6.776, with measured values given on the total scale ($pH_{tot}$).

For TA measurements, 40 g of seawater were titrated with 0.1 N HCl at 25°C.

Titration continued until the pH exceeded the end point (~pH 4.4), and then continued
until ~pH 3.0, with the potential change and titration volume recorded. The consumed
volume of HCl was calculated using the Gran (1952) function based on the linear
relationship between titration volume and pH, and TA was obtained by plotting the
consumed volume of HCl. The reference material for experimental quality control
was obtained from Professor Andrew Dickson (Scripps Institute of Oceanography,
USA), and the pH of the reference material was calculated by entering dissolved
inorganic carbon (DIC) and TA data into $CO_2SYS$ (Lewis and Wallace, 1998).





The pH$_{NBS}$ measurement accuracy in this study was ±0.01 units, the pH$_{tot}$
accuracy was ±0.009 units, and the TA accuracy was ±2.7 µmol kg$^{-1}$
(precision=0.12%). $p$CO$_2$ was also calculated with CO$_2$SYS from measured pH and
TA. The dissociation constants of carbonic acid used were the revised K1 and K2
values from Mehrbach et al. (1973) and Dickson and Millero (1987).
**2.2 Calculation of the exchange flux of CO$_2$ between the ocean and the**
**atmosphere** The formula for calculating the exchange flux of CO$_2$ between the ocean
and the atmosphere was as follows:
F$_{GAS}$=k × $K_H$ × ($p$CO$_2$seawater−$p$CO$_2$air)
where k is the gas exchange rate of CO$_2$ (air-sea gas transfer rate), which was
obtained from an empirical formula based on wind speed proposed by Wanninkhof
(1992): k=0.31 × u2 × ($Sc$/660)-0.5, where $u$ is wind speed 10 m above sea level (in
m/s; data from the Central Weather Bureau's Oceanic Center-Erluanbi buoy); $Sc$
(Schmidt number) is a function of temperature, which can be obtained from the $in situ$
sea surface temperature (T) as follows:
$Sc$ = 2073.1 − 125.62 × T + 3.6276 × T2 − 0.043219 × T3
The solubility of CO$_2$ gas in seawater ($K_H$), expressed in moles/L·atm, was calculated
using the formula developed by Weiss (1974):
$$\ln K_H = -58.0931 + 90.5069\left(\frac{100}{T}\right) + 22.2940\ln\left(\frac{T}{100}\right) + S\left[0.027766 - 0.025888 + \left(\frac{T}{100}\right) + 0.0050578\left(\frac{T}{100}\right)^2\right]$$



Since we did not measure atmospheric $CO_2$, we used data from air samples collected
by the United States National Oceanic and Atmospheric Administration (NOAA) at
Dongsha Island: 397 µatm in March 2011, 392 µatm in July 2011, 392 µatm in
October 2011, and 396 µatm in January 2013
(https://gml.noaa.gov/dv/data/index.php?category=Greenhouse%2BGases¶meter
_name=Carbon%2BDioxide&site=DSI).

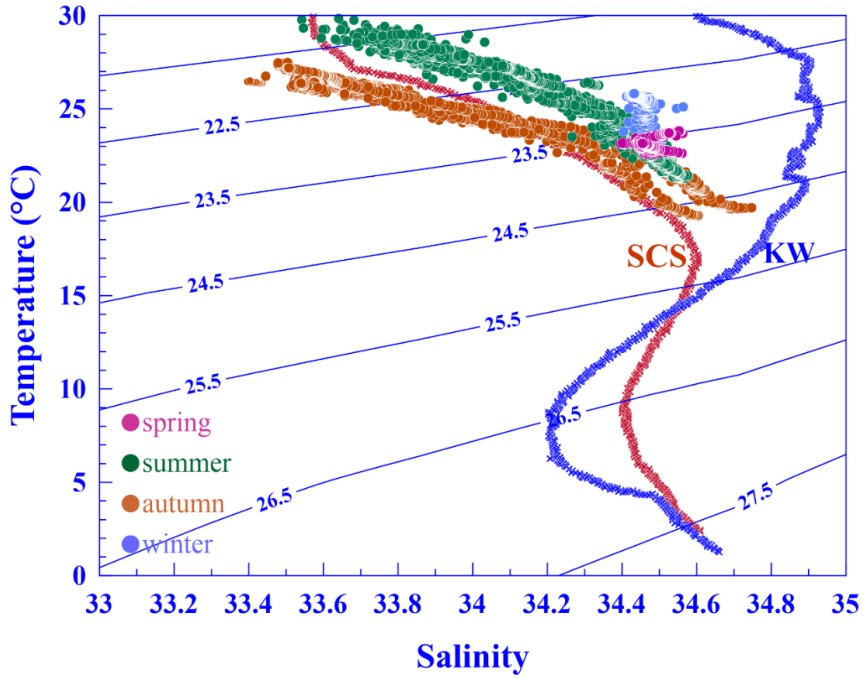


**Fig. 2** Temperature vs. salinity (T-S) diagram at Nanwan Bay, Taiwan in spring,
summer, autumn, and winter. SCS = South China Sea and KW = Kuroshio
Current waters.

3.  **Results and discussion**



**3.1 Variation in hydrological parameters** Both temperature and salinity varied
over time in Nanwan Bay (Fig. 2), with the seasonal variation likely driven by both
the monsoon and SCS circulation patterns. The Kuroshio Current flows northward
along Taiwan's east coast, with a portion of Western Philippine Sea (WPS) water
following the Kuroshio and then flowing westward along the northern SCS shelf
(Yuan et al., 2006). Nan et al. (2015) suggested that surface salinity of 34 or higher is
characteristic of the Kuroshio, indicating potential inundation of the Kuroshio Current
into Nanwan Bay during high-salinity spring periods. During summer, the southwest
monsoon dominates, leading to a decrease in the Kuroshio's influence; the main
circulation of the Kuroshio shifts westward to the Luzon Strait, limiting its intrusion
into the northwestern SCS region (Liang et al., 2008). In the northern SCS region, the
southwest-to-northeast circulation prevails during the monsoon, with most seawater
flowing out of the SCS through the Luzon Strait and converging with the Kuroshio
axis, resulting in Nanwan Bay being dominated by the SCS water mass during the
summer. Analysis of temperature and salinity data from Nanwan Bay, the SCS, and
the Kuroshio current indicates that Nanwan Bay mainly consists of the SCS water
mass during summer and autumn, while during spring and winter, the water masses
are intermediate between the two (Fig. 2). As such, Nanwan Bay is classified as a
mixed water mass area, comprising both SCS and Kuroshio Current water masses.



**Table 1.** Correlation matrix of variables with correlation coefficient ($r$) for spring,
summer, autumn, and winter. Variables include temperature, salinity,
dissolved oxygen (DO), saturation of DO (DO%), total alkalinity (TA),
dissolved inorganic carbon (DIC), and pH.

| **Spring** | Temperature | Salinity | DO | DO (%) | TA | DIC | pH |
|---|---|---|---|---|---|---|---|
| Salinity | 0.35** | | | | | | |
| DO | -0.23 | -0.40** | | | | | |
| DO (%) | 0.02 | -0.32** | 0.89** | | | | |
| TA | 0.04 | 0.03 | 0.19 | 0.11 | | | |
| DIC | -0.17 | -0.04 | 0.25* | 0.11 | 0.92** | | |
| pH | 0.43** | 0.13 | -0.02 | 0.05 | 0.61** | 0.27* | |
| $p$CO$_2$ | -0.28* | -0.07 | 0.03 | -0.02 | -0.45** | -0.09 | -0.96** |


| **Summer** | Temperature | Salinity | DO | DO (%) | TA | DIC | pH |
|---|---|---|---|---|---|---|---|
| Salinity | -0.96** | | | | | | |
| DO | 0.65** | -0.53** | | | | | |
| DO (%) | 0.90** | -0.81** | 0.90** | | | | |
| TA | -0.82** | 0.81** | -0.52** | -0.72** | | | |
| DIC | -0.91** | 0.84** | -0.68** | -0.87** | 0.91** | | |
| pH | 0.89** | -0.79** | 0.72** | 0.89** | -0.72** | -0.94** | |
| $p$CO$_2$ | -0.22* | 0.07 | -0.45** | -0.38** | 0.23* | 0.51** | -0.64** |


| **Autumn** | Temperature | Salinity | DO | DO (%) | TA | DIC | pH |
|---|---|---|---|---|---|---|---|
| Salinity | -0.95** | | | | | | |
| DO | 0.88** | -0.85** | | | | | |
| DO (%) | 0.95** | -0.93** | 0.98** | | | | |
| TA | -0.57** | 0.56** | -0.51** | -0.55** | | | |
| DIC | -0.76** | 0.75** | -0.66** | -0.72** | 0.88** | | |
| pH | 0.79** | -0.77** | 0.66** | 0.73** | -0.43** | -0.80** | |
| $p$CO$_2$ | -0.32** | 0.33** | -0.22 | -0.27* | 0.27* | 0.63** | -0.83** |


| **Winter** | Temperature | Salinity | DO | DO (%) | TA | DIC | pH |
|---|---|---|---|---|---|---|---|
| Salinity | -0.32** | | | | | | |
| DO | 0.43** | -0.26* | | | | | |
| DO (%) | 0.78** | -0.34** | 0.70** | | | | |
| TA | -0.15 | 0.06 | -0.19 | -0.17 | | | |
| DIC | -0.39** | 0.02 | -0.34** | -0.41** | 0.89** | | |
| pH | 0.59** | 0.06 | 0.39** | 0.61** | -0.12 | -0.56** | |
| $p$CO$_2$ | -0.34** | -0.19 | -0.33** | -0.44** | 0.29** | 0.67** | -0.94** |

*: $p \leq 0.05$ and **: $p \leq 0.01$.



During the survey period, there was a clear positive correlation between pH and
temperature. Additionally, pH and TA exhibited significant correlations with salinity
during summer and autumn, but not in spring and winter (Table 1). These findings
suggest that vertical mixing occurs in spring and winter, whereas the positive
correlation between TA and pH in the spring season indicates upwelling. It is expected
that TA and salinity will covary because the charge differences between cations and
anions in seawater change with salinity. Salinity generally increases with depth and is
influenced by various factors such as rainfall, evaporation, and freshwater input,
which can lead to changes in TA.

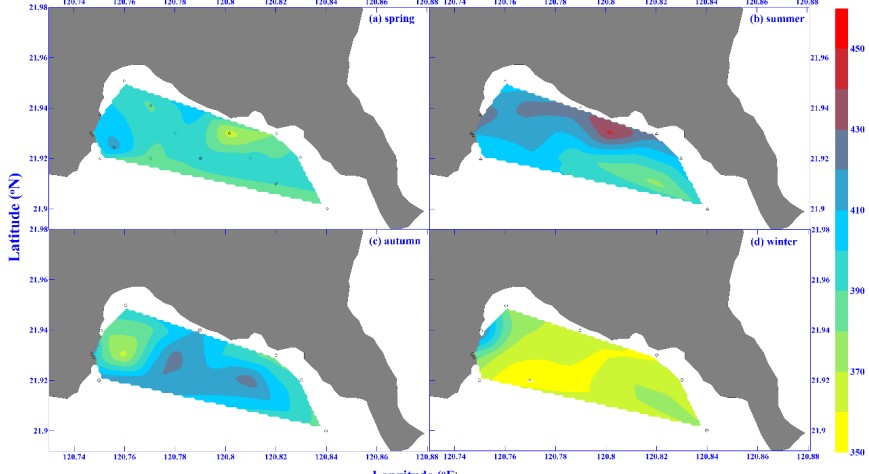


**Fig. 3** Seasonal variation in sea surface $p\mathrm{CO}_2$ (µatm) in Nanwan Bay, Taiwan in

spring (a), summer (b), autumn (c), and winter (d). Values in legends along the

right sides of panels correspond to µatm.

**3.2 Changes in surface water $p\mathrm{CO}_2$** $p\mathrm{CO}_2$ levels in Nanwan Bay ranged from





364–422 µatm, 362–448, 350–480, and 345–427 µatm in spring, summer, autumn,
and winter, respectively (Fig. 3). The means (±SD) for these levels were 393.2
(±11.6), 411.4 (±19.0), 401.7 (±18.3), and 370 (±17.3) µatm, respectively. The mean
temperatures during these seasons were 23.3, 28.5, 26.9, and 26℃, respectively. In the
open ocean, $p$CO$_2$ levels are primarily influenced by temperature, horizontal transport
and vertical mixing, biological processes, and gas exchange (e.g., Dai et al., 2009).

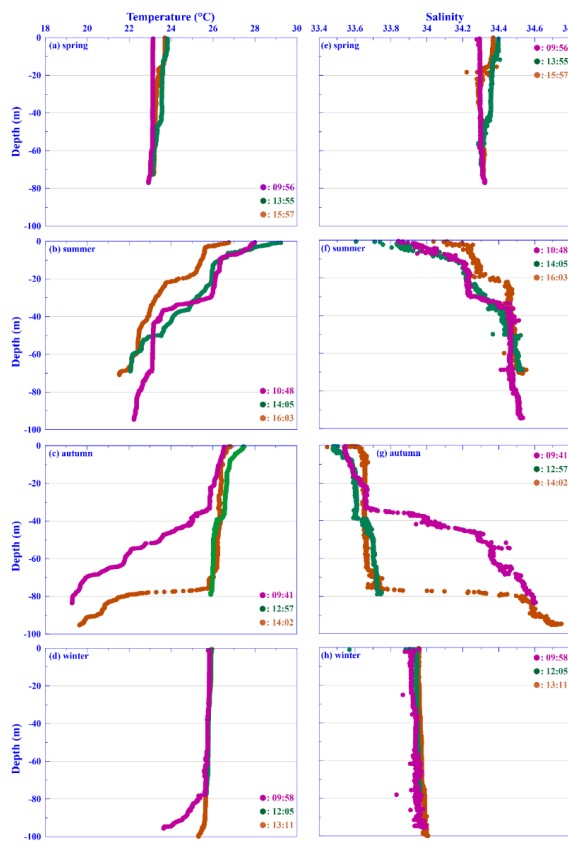


**Fig. 4** Vertical profiles of temperature and salinity at station S10 in spring (a & e,

respectively), summer (b & f, respectively), autumn (c & g, respectively), and

winter (d & h, respectively) at three sampling times as indicated in each panel.

Due to the mixing of different water masses by monsoons, tides, eddies, upwelling,
and other ocean currents, significant gradient changes were observed at different
times at station S10 (Fig. 4) and in the carbonate parameter data (Figs. 5–8). In
summer and autumn, more obvious mixing occurred, as seen by the vertical variation
in temperature and salinity, while in spring and winter, mixing was less evident.

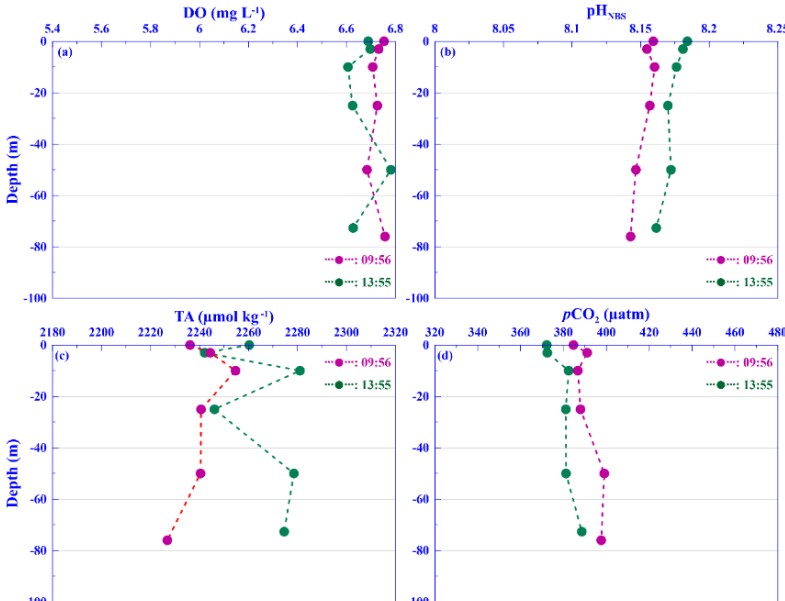


**Fig. 5** Vertical profiles of dissolved oxygen (DO; a), pH (b), total alkalinity (TA; c),

and $p$CO$_2$ (d) in spring at station S10 at two sampling times.

According to Lee et al. (1997; 1999a; 1999b), cold-water upwelling occurs with tidal
changes in Nanwan Bay, which increases vertical mixing. The temperature-salinity-
pH-DO diagram of station S1 shows that during cold-water upwelling, cold, low-DO,
low-pH, and high-salinity deep-sea water intrudes the nearshore regions of Nanwan





Bay (Fig. 9). Seawater quality profiles of S10 provide further evidence of upwelling,
such as low temperatures, low DO, low pH, high salinity, and $pCO_2$ increases of 31
and 37 µatm in summer and autumn, respectively (Figs. 4 & 6–7).

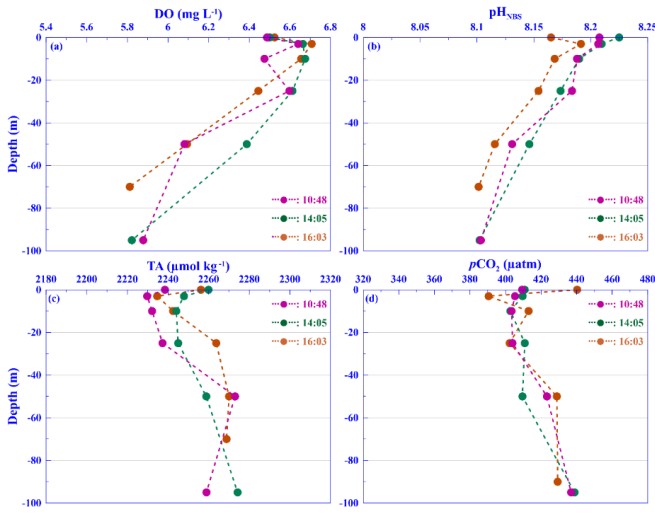


**Fig. 6** Vertical profiles of dissolved oxygen (DO; a), pH (b), total alkalinity (TA; c),

and $pCO_2$ (d) in summer at station S10 at three sampling times.

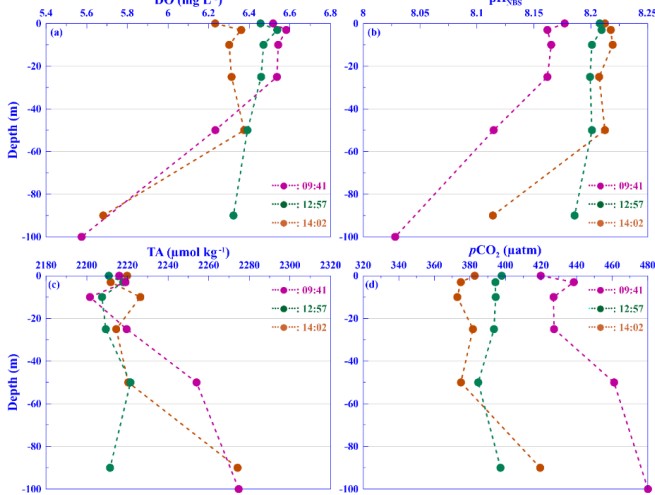


**Fig. 7** Vertical profiles of dissolved oxygen (DO; a), pH (b), total alkalinity (TA; c),

and $pCO_2$ (d) in autumn at station S10 at three sampling times.

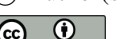

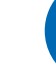

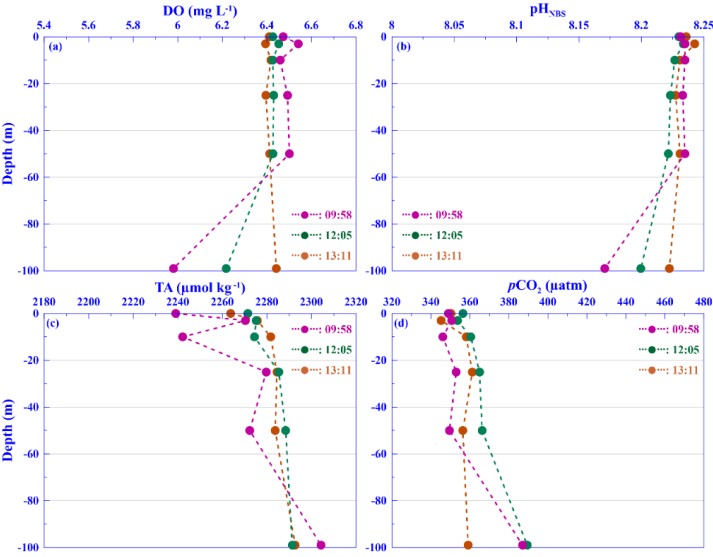


**Fig. 8** Vertical profiles of dissolved oxygen (DO; a), pH (b), total alkalinity (TA; c),

and $p\mathrm{CO_2}$ (d) in winter at station S10 at three sampling times.

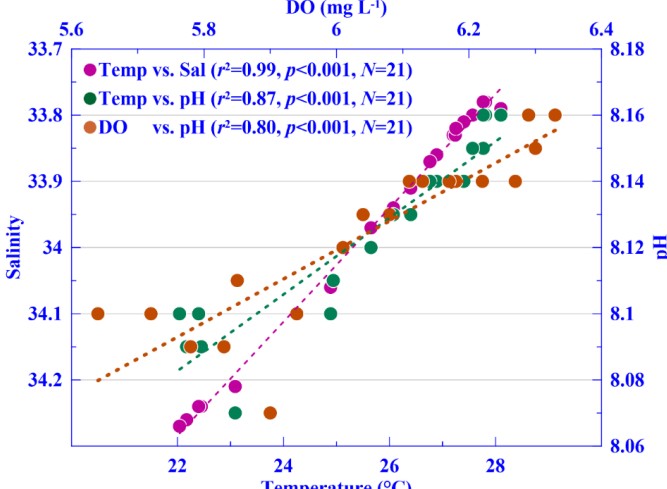


**Fig. 9** Relationships between temperature and salinity or pH, as well as dissolved

oxygen (DO) vs. pH during upwelling events (data from Tew et al., 2014).
As temperature increases, $\mathrm{CO_2}$ solubility decreases, causing an increase in $p\mathrm{CO_2}$.
Takahashi et al. (2002) proposed to evaluate the relative effects of temperature and



non-temperature effects on $p\mathrm{CO_2}$ changes as follows:

$p\mathrm{CO_2}$ at $\mathrm{T_{obs}} = (p\mathrm{CO_2})_{\text{Mean annual}} \times \exp[0.0423(\mathrm{T_{obs}} - \mathrm{T_{mean}})]$

$p\mathrm{CO_2}$ at $\mathrm{T_{mean}} = (p\mathrm{CO_2})_{\text{obs}} \times \exp[0.0423(\mathrm{T_{mean}} - \mathrm{T_{obs}})]$

$p\mathrm{CO_2}$ at $\mathrm{T_{obs}}$ is calculated using the average $p\mathrm{CO_2}$ to determine the $p\mathrm{CO_2}$ value at the
measured temperature, assuming that the change in $p\mathrm{CO_2}$ is due to temperature; $p\mathrm{CO_2}$
at $\mathrm{T_{mean}}$ is the standardized $p\mathrm{CO_2}$ value at the average temperature, assuming that the
change in $p\mathrm{CO_2}$ is *not* due to temperature; $\mathrm{T_{mean}}$ and $\mathrm{T_{obs}}$ are the annual average
temperature and the measured temperature on-site, respectively.

The mean $p\mathrm{CO_2}$ of the monitoring stations varied over time, indicating that

temperature and non-temperature effects had different impacts on the average $p\mathrm{CO_2}$ of
each station (Fig. 10). This suggests that seasonal changes in $p\mathrm{CO_2}$ are influenced by
both temperature and non-temperature effects, with some stations showing larger
changes than others. It is believed that the stations with larger $p\mathrm{CO_2}$ changes are
primarily affected by one type of effect (temperature vs. other), while the smaller
changes reflect the mutual offsetting of the two effects. The variability in $p\mathrm{CO_2}$
observed at S31 and S33, which are located near the Nuclear Power Plant outlet, is
likely driven by temperature change, as the water temperature in this area is
consistently higher than that of the surrounding area throughout the year. In fact, we
expected that temperature effects on $p\mathrm{CO_2}$ would be more pronounced at these sites.



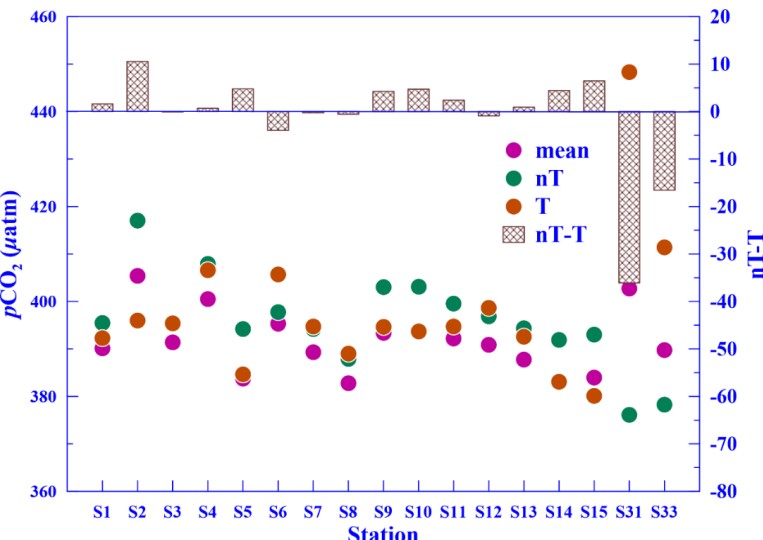


**Fig. 10** Control factors and impact levels of surface water $pCO_2$ at each station in

Nanwan Bay. "Mean" refer to average value of each station, "nT" refers to non-

temperature effects, "T" refers to temperature effects, and "nT-T" represents the

degree of influence.

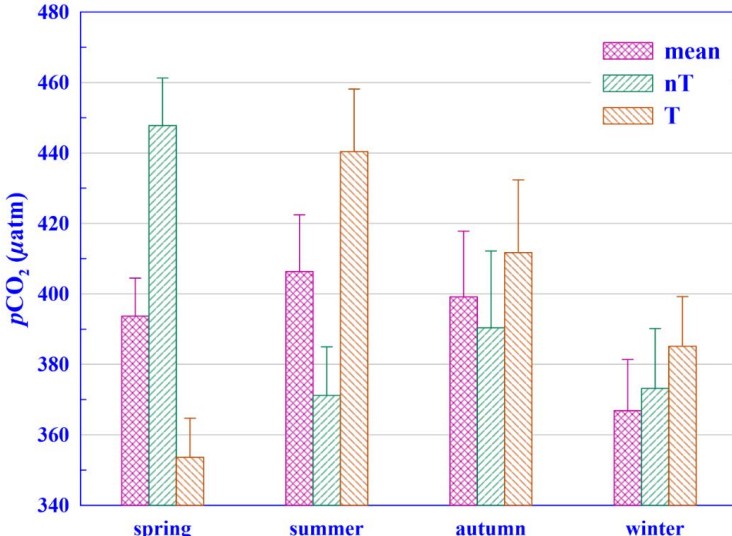

**Fig. 11** Control factors of surface water $pCO_2$ in different seasons in Nanwan Bay.

Abbreviations are as in Fig. 10. The standard deviations are shown as vertical

lines.



During the investigation period, seasonal variation in surface water $pCO_2$ in

Nanwan Bay was mainly affected by non-temperature effects in the spring,
temperature in the summer, and both effects in the autumn and winter (Fig. 11). The
relationship between $pCO_2$, surface water temperature, and Chl $a$ concentration
revealed a significant correlation between $pCO_2$ and temperature in the summer
($p<0.01$; Fig. 12a), and a positive correlation between $pCO_2$ and Chl $a$ in autumn
($p<0.05$; Fig. 12b). This suggests that temperature and Chl $a$ are the main factors
affecting $pCO_2$ in summer and autumn, respectively.

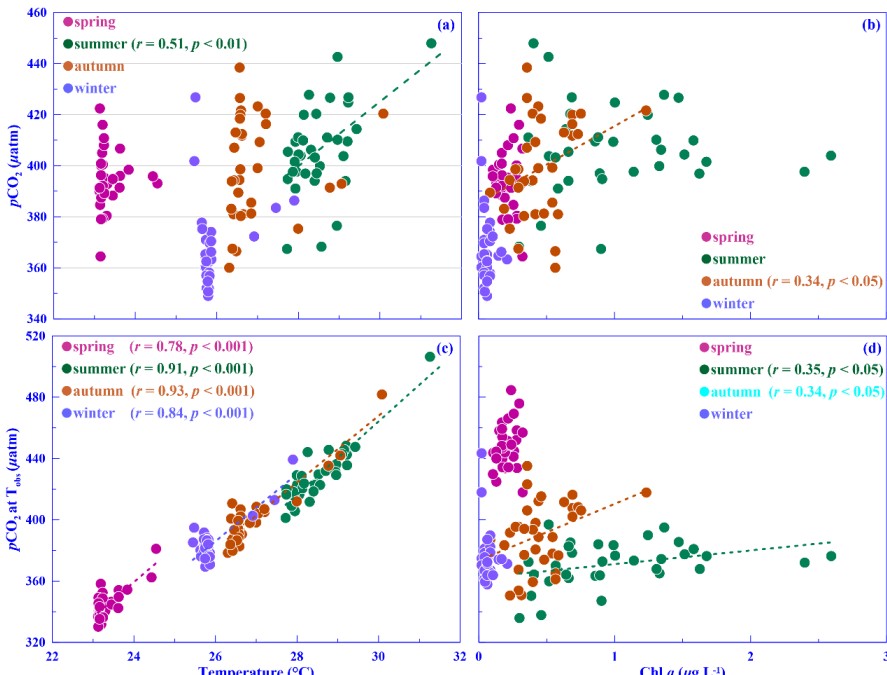


**Fig. 12** Relationship between surface water $pCO_2$ and temperature (a), surface water

$pCO_2$ and Chl $a$ (b), surface water $pCO_2$ at $T_{obs}$ and temperature (c), and surface

water $pCO_2$ at $T_{mean}$ and Chl $a$ (d) in different seasons in Nanwan Bay.



To evaluate the impact of temperature and non-temperature effects on seasonal
variation in $p$CO$_2$, we compared the average $p$CO$_2$ under actual temperature
conditions with $p$CO$_2$ values standardized to mean temperature (i.e., excluding the
temperature effect). We found a significant correlation between $p$CO$_2$ and actual
temperature (Fig. 12c), and a significant positive correlation between $p$CO$_2$ and Chl $a$
in summer and autumn, when temperature effects were excluded (Fig. 12d; all
$p<0.05$). This suggests that temperature is the main factor driving seasonal variation
in $p$CO$_2$, although non-temperature effects, specifically Chl $a$, also have an influence.
However, given the low $r$ values, there are likely unmeasured parameters that
contribute to the variation in $p$CO$_2$ over time.
Chen et al. (2004a) proposed the Degree of Nutrient Consumption (DNC) as an
indicator of upwelling magnitude. This indicator is characterized by changes in
temperature, DO, pH, salinity, alkalinity, DIC, nutrients, and Chl $a$ during an
upwelling event. These changes can alter the $p$CO$_2$ of surface waters through
chemical and biological processes (Chen and Hsing, 2005).
Nanwan Bay's benthic environmental system has a regenerative effect. However,
due to the high shallow water temperature and frequent stratification, regenerative
nutrients cannot easily be transported to the shallows. This results in the shallow areas
rarely becoming eutrophic (Leichter et al., 1996; Torréton, 1999; Wolanski and





Pickard, 1983). Another reason for Nanwan Bay's oligotrophy is that when nutrients
flow into reef areas, resident organisms quickly utilize them. Although nutrient input
from outside the bay is greater than the outward flux (Su, 2009), rapid circulation of
water in Nanwan Bay leads to unused nutrients being swiftly exported out of the bay.
This causes oligotrophy and high benthic productivity in the area. Su (2009) reports
that during spring tides, the water can be replaced in just 1.6 tidal cycles. Therefore,
nutrient levels and Chl *a* may have only small influences on $p\text{CO}_2$ in Nanwan Bay,
with temperature changes and seawater movement having a more significant impact.
**3.3 Spatial distribution of $\triangle p\text{CO}_2$ and $\text{CO}_2$ air-sea flux** The partial pressure
difference between $\text{CO}_2$ in surface seawater and the atmosphere, denoted as $\triangle p\text{CO}_2$,
indicates the direction of air-sea $\text{CO}_2$ exchange. When $\triangle p\text{CO}_2 > 0$, the seawater is
supersaturated with $\text{CO}_2$ and releases it into the atmosphere, contributing to an
increase in atmospheric $\text{CO}_2$ concentration (i.e., a source). On the other hand, when
$\triangle p\text{CO}_2 < 0$, $\text{CO}_2$ from the atmosphere enters the seawater, acting as a sink for
atmospheric $\text{CO}_2$. In spring, the $\triangle p\text{CO}_2$ range was between -24 and 18 µatm, with an
average of -2.3 µatm (Fig. 13a). The highest value was observed near the Nuclear
Power Plant outlet station. In summer, it ranged between -20 and 39 µatm, with an
average of 14.3 µatm. The highest value was measured near station S7. In autumn, the
range was between -29 and 37 µatm, with an average of 7.2 µatm. The highest value





was observed near stations S10-S12. In winter, the range was between -46 and 18
µatm, with an average of -29 µatm. The lowest value occurred near stations S7-S10.

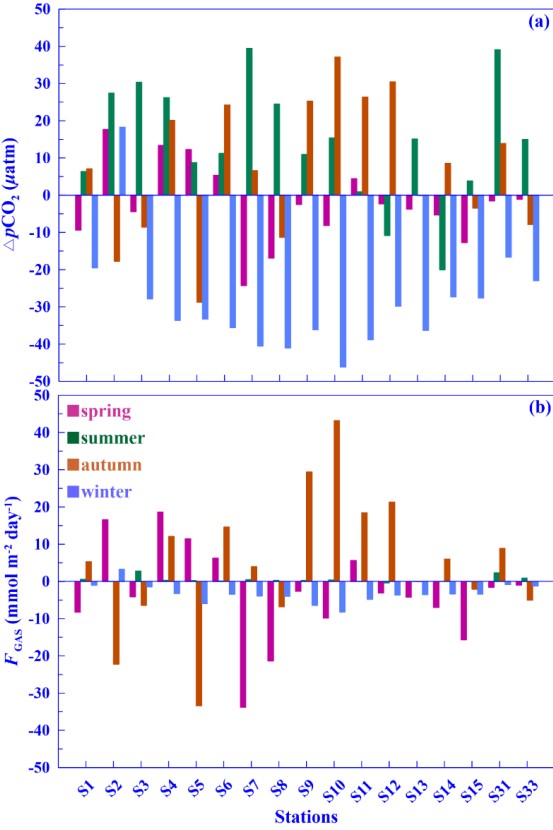


**Fig. 13** Seasonal variation of (a) surface water $\triangle pCO_2$ and (b) air-sea $CO_2$ exchange

flux ($F_{GAS}$) at each station. Values are presented relative to the annual mean

(scaled to 0).

Based on data from the Central Weather Bureau-Guanyinshan buoy, the average

wind speed during the southwest monsoon season (summer) was 1.67 m s$^{-1}$, while
during the northeast monsoon seasons (spring, autumn, & winter), it was 10, 8.1, and
2.8 m s$^{-1}$, respectively. Using these values, the $CO_2$ air-sea exchange flux in Nanwan



Bay was calculated (Figure 13b). During spring, the $CO_2$ flux ranged from -33.8 to
18.7 mmol m$^{-2}$ day$^{-1}$ (average=-3.2 mmol m$^{-2}$ day$^{-1}$). In summer, the $CO_2$ flux ranged
from -0.4 to 2.8 mmol m$^{-2}$ day$^{-1}$ (average=0.5 mmol m$^{-2}$ day$^{-1}$). During autumn, the
$CO_2$ flux ranged from -33.4 to 43.2 mmol m$^{-2}$ day$^{-1}$, with an average of 5.1 mmol m$^{-2}$
day$^{-1}$. Finally, in winter, the $CO_2$ flux ranged from -8.2 to 3.3 mmol m$^{-2}$ day$^{-1}$, with an
average of -3.3 mmol m$^{-2}$ day$^{-1}$. These findings demonstrate that wind speed is a
crucial factor affecting $CO_2$ air-sea exchange flux.

The area between Cape Maobitou and Cape Oluanpi, covering approximately 30

km$^2$ (Fig. 1), showed an annual absorption of approximately -29 tC, with seasonal
fluxes of -106 tC, 16 tC, 167 tC, and -107 tC in spring, summer, fall, and winter,
respectively. This study's calculated value of ~-1 gC m$^{-2}$ year$^{-1}$ as a net sink in
Nanwan Bay contrasts with $CO_2$ sea-air flux data from other coral reef areas, such as
Bermuda (14.4 gC m$^{-2}$ year$^{-1}$; Bates et al., 2001), Okinawa (21.6 gC m$^{-2}$ year$^{-1}$; Ohde
and Van Woesik, 1999), the Great Barrier Reef (18 gC m$^{-2}$ year$^{-1}$; Frankignoulle et
al., 1996), French Polynesia (1.2 gC m$^{-2}$ year$^{-1}$; Frankignoulle et al., 1996; Gattuso et
al., 1997; Gattuso et al., 1993), and Hawaii (17.4 gC m$^{-2}$ year$^{-1}$; Fagan and
Mackenzie, 2007), all atmospheric $CO_2$ sources. The biogeochemistry of nearshore
environments is impacted by land-based inputs, resulting in differences in $p$$CO_2$ and
variations in the $CO_2$ sea-air flux. Prior studies showed that ship-based and satellite-



based wind field calculations of the $CO_2$ sea-air flux in the East China Sea have
similar trends, but there are significant differences in absolute values, with ship-based
wind field calculations producing greater $CO_2$ exchange than satellite-based
calculations (Tseng et al., 2011). Short-term and long-term wind speeds, climate
conditions, and specific events can cause differences in $CO_2$ flux calculations due to
the nonlinear empirical relationship between wind speed and gas exchange, leading to
varying results (Chou et al., 2011; Evans et al., 2012; De La Paz et al., 2011).

**4.  Conclusions**

Nanwan Bay is a region that experiences significant changes in temperature and

salinity throughout the year, primarily influenced by the SCS and the Kuroshio
Current. These seasonal variations have an impact on the seawater carbonate system
(including $p$CO$_2$), which can also be influenced by vertical water movement and
biological activity. Temperature was the most important driver of spatio-temporal
differences in $p$CO$_2$, particularly at the consistently warmer outlet station. However,
non-temperature effects also played a role in the spring, while the interaction between
temperature and other factors was important in the autumn and winter. In terms of its
impact on atmospheric $CO_2$ levels, Nanwan Bay acts as a sink in the spring and
winter. However, in the summer and autumn, particularly during upwelling events, it
becomes a source of atmospheric $CO_2$, releasing more $CO_2$ than it absorbs. Overall,





395 the complex interplay of temperature, water mass origin, vertical water movement,

396 and biological activity in Nanwan Bay has a significant impact on its carbon dioxide

397 dynamics and its influence on atmospheric $CO_2$ levels.


399        **5. Acknowledgements**

400  This study was supported by grants from the Ministry of Science and Technology

401 of Taiwan (MOST 111-2611-M-259-002, MOST 110-2611-M-259-002, MOST 109-

402 2611-M-259-003, MOST 108-2611-M-291-005, MOST 107-2611-M-291-001, &

403 MOST 106-2611-M-291-006) to PJM and CCC (MOST 111-2611-M-003-005).

404 Data were submitted to Dryad for archiving (doi: 10.5061/dryad.63xsj3v7d).

406        **6. Credit author statement**

407  This manuscript was conceptualized by PJM and CCC; CMC and HYH conducted

408 investigations on all cruises and collected and analyzed the initial data; PJM, ABM,

409 and CCC wrote the initial draft; all authors provided comments and edits. The authors

410 declare that they have no conflict of interest.



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
