# Peer review of "The impact of coral reef ecosystems and upwelling events on the marine carbon dynamics of Southern Taiwan"

_EGUsphere, 2023_

## Author Comment (AC1)

**Response to reviewers' 1 comments on ms no: egusphere-2023-1097 "The impact of coral reef ecosystems and upwelling events on the marine carbon dynamics of Southern Taiwan" (Meng, Chang, Hsieh, Mayfield, and Chen)**

1. The manuscript discusses the carbonate system measurements in Nanwan Bay, Taiwan, and in particular focuses on the processes that influence $pCO_2$ in Nanwan Bay and seasonal variability in whether the bay is a source or a sink.
   I have concerns about the methods used in this study. The study uses the gas exchange relationship $FGAS = k \times K_H \times (pCO_{2,seawater} - pCO_{2,air})$, and so conclusions about the magnitude of the air-sea gas exchange are highly dependent on $pCO_{2,seawater}$, $pCO_{2,air}$, and the gas exchange rate k which is in turn highly dependent on wind speed. I believe that there are serious issues that need to be addressed with each of these parameters.

   **Response: We deeply appreciate the thorough review you conducted on our manuscript. In response to your insightful recommendations and those of other reviewers, we have undertaken substantial revisions. Your valuable and constructive comments have not only elevated the quality of our manuscript but have also motivated us to reconsider the presentation and interpretation of our results.**

   **For your convenience, we have provided detailed responses to your comments below, with a specific focus on your concerns regarding the uncertainty of the $pCO_2$ estimation and sea-air gas-exchange. These concerns have been thoroughly addressed in our responses to your comments 1-7. We hope that the level of detail in our responses reflects our dedication to incorporating your feedback into this revised manuscript, and, overall, we are confident that we have addressed these comments in a clear and reasonable manner to where, in the end, a manuscript of superior quality has emerged. We are now confident that this manuscript fully meets the publication standards set by _Ocean Science_.**

$pCO_{2,seawater}$

2. The $pCO_{2,seawater}$ values used in this study were calculated from pH and TA measurements, however these data are not presented. pH measurements were recorded on two different scales, and it was not explained why, which measurements were made on which scale, or how this may have influenced any of the results.

   **Response: Thank you for your valuable suggestion. In this study, we calculated the $pCO_2$ in seawater indirectly: by using measured values of pH and TA. It is worth noting that we employed the $pH_{NBS}$ scale for this calculation. We prepared pH buffers with a value of 8.083 for the Tris artificial seawater buffer (2-amino-2-hydroxymethyl-1,3-propanediol) and a value of 6.776 for the AMP**

artificial seawater buffer (2-aminopyridine) to calibrate the electrode for measuring $pH_{tot}$. For additional details, please refer to our response to your comment #22.

To enhance clarity, we have opted to exclude the use of $pH_{tot}$ throughout this manuscript, relying exclusively on $pH_{NBS}$ in this revised version. Furthermore, we have included some pH and TA values in Fig. 5 for comparative purposes. We hope this addresses your concerns related to this matter.

3. It is not clear how the values obtained during the study were averaged to obtain the mean $pCO_{2,seawater}$ values for each site. This is problematic because the sites are not evenly spaced, and so a simple arithmetic mean would result in certain areas (ie., around the nuclear power plant outflow) being overweighted, while others would be underweighted.

Response: Thank you for bringing up the issue of unevenly spaced sampling, particularly in relation to the stations located around the nuclear power plant outflow. To assess the potential impact of this issue, we conducted an evaluation of mean values and considered both temperature and non-temperature effects on surface water $pCO_2$. This assessment was done with and without the data from the stations surrounding the nuclear power plant outflow (S31 and S33). The results have been graphically presented for your reference, with Panel (a) displaying the analysis including the data from these stations and Panel (b) excluding them (data are also provided in the table for reference).

Based on our findings, it appears that the unevenly spaced sampling does not significantly affect the estimation of mean values. This is likely attributed to the fact that there were only 2 data points out of a total of 17 sampling stations. Furthermore, the inclusion of these two stations (S31 and S33) in our analysis serves as a valuable cross-reference, reaffirming the accuracy of our measurements. As a result, we have chosen to retain data from both stations (S31 and S33) for our estimation in this revised analysis.

[Figure]

Fig. R1. Mean values and impact levels of surface water $pCO_2$ in Nanwan Bay during

**different seasons with (a) and (b) without (b) values measured from S31 and S33.**

**Table R1. Data for Fig. R1. SD=standard deviation.**

|  | Seasons | Mean | nT | T | mean SD | nT SD | T SD |
|---|---|---|---|---|---|---|---|
| With | spring | 393.7 | 49.4 | -103.5 | 11.55 | 11.45 | 16.87 |
| S31 and | summer | 406.3 | -22.4 | 57.5 | 17.18 | 14.85 | 20.60 |
| S33 | autumn | 399.2 | -1.7 | 10.5 | 19.29 | 24.05 | 34.29 |
| data | winter | 366.9 | -18.3 | 12.0 | 15.22 | 17.20 | 25.27 |
| Without | spring | 393.6 | 46.2 | -95.2 | 11.55 | 10.63 | 11.18 |
| S31 and | summer | 404.6 | -26.9 | 65.2 | 15.78 | 14.27 | 15.05 |
| S33 | autumn | 399.7 | -1.5 | 12.1 | 19.39 | 17.31 | 17.05 |
| data | winter | 364.6 | -22.6 | 19.4 | 10.81 | 12.74 | 14.56 |

4. It is also stated that measurements were taken on particular days, but it does not say how many times these measurements were taken. It appears that there were at least three measurements at S10 during many of the cruises, with these three measurements sometimes showing distinct variations in surface $pCO_2$ of up to ~ 50 µatm during the same day, and greater than this at depth. These large variations call into question how representative the values presented are of the system as a whole, and some discussion is needed to justify why the numbers here would be representative.

> **Response: Thank you for bringing to our attention the unclear sampling scheme used in this study. In response to your valuable input, we have now included detailed information regarding the sampling time and depth at each station for different seasons in Table S1.**
>
> **To investigate diurnal variations in this coral reef ecosystem, we selected S10, one of the 17 sampling stations, for a more in-depth analysis. At S10, we conducted three separate sampling events on each sampling date during various seasons. Notably, we observed significant diurnal variations at S10, which aligns with established patterns found in many coastal ecosystems, including coral reefs.**
>
> **In order to facilitate spatial and seasonal comparisons, we limited our analysis at S10 to data collected during time periods closely matching those of other stations. This approach allowed us to calculate more accurate mean values while minimizing the influence of diurnal variations originating specifically from station S10. Hopefully, you will feel that this is a reasonable approach.**

$pCO_{2,air}$

5. This study uses $pCO_{2,air}$ values measured at Dongsha Island, a remote island ~500km

away from Taiwan and 250 km away from the nearest landmass. As such, I think there needs to be some justification as to why these values are representative of Nanwan Bay. Could there be anthropogenic influences to the pCO₂ near Nanwan Bay that do not influence Dongsha Island? Could the seasonal terrestrial signal have a greater impact on Nanwan Bay than Dongsha Island? A sensitivity analysis to determine how much influence variations in $pCO_{2,air}$ values would have on the calculated fluxes would be beneficial.

**Response: Thank you for highlighting this crucial matter, which was actually raised by the other reviewer, as well. In this study, we acquired $pCO_2^{air}$ data for Dongsha Island from the NOAA website. Dongsha Island is one of two $pCO_2^{air}$ monitoring stations in Taiwan, with the other station situated at Lulin, a high-altitude location (2,862 m) within the Taiwanese mountains (refer to the table below for details.). For your reference, the table also presents $pCO_2^{air}$ values of these two stations obtained on similar sampling dates during this study.**

**It is evident from the data that the $pCO_2^{air}$ values at both stations were similar, except during the summer months. This discrepancy in summer values may be attributed to the robust growth of terrestrial vegetation in the vicinity of the Lulin station. However, we maintain that this seasonal terrestrial influence likely has only a minor impact on Nanwan Bay, particularly considering the persistence of the southwest monsoon during the summer season.**

**Similar to Nanwan Bay, both $pCO_2^{air}$ monitoring stations were strategically located in remote areas, far removed from urban centers and industrial zones, to minimize anthropogenic interference. To estimate the $CO_2$ exchange flux between the ocean and the atmosphere, we opted to utilize the $pCO_2^{air}$ data from Dongsha Island. This choice is justified by similar latitude to Nanwan Bay (approximately 21.90°N) and, importantly, the shared characteristic of being part of a coral reef ecosystem (Dongsha is a coral atoll.).**

**Therefore, we are confident that the $pCO_2^{air}$ values observed at Dongsha Island are suitable for our flux estimation, despite the station's geographical distance from our study area.**

**Table R2. The $pCO_2^{air}$ values on the similar sampling dates of those of our study in two stations in Taiwan (from NOAA, USA).**

| Stations | Dongsha Island | Lulin |
|---|---|---|
| Site latitude (ºN) | 20.6992 | 23.47 |
| Site longitude (ºE) | 116.7297 | 120.87 |
| Site elevation (m) | 3.0 | 2862.0 |
| $pCO_2^{air}$ (03/29/2011) | 397 µatm | 394 µatm |

| | | |
|---|---|---|
| $p$CO$_2$$^{air}$ (07/05/2011) | 392 µatm | 385 µatm (07/06/2011) |
| $p$CO$_2$$^{air}$ (10/18/2011) | 392 µatm | 392 µatm |
| $p$CO$_2$$^{air}$ (01/22/2013) | 396 µatm | 397 µatm |

Wind speed

6. This study uses the seasonal average wind speed to calculate the seasonal air-sea gas exchange for Nanwan Bay, however I think there needs to be more justification about why this is reasonable. The authors formulation of k is quadratically dependent on wind speed, and so a few relatively brief high wind speed events could result in the gas exchange being underestimated if only the average wind speed is used. Data showing that the wind speed in the bay is highly consistent, or some work to characterize how such occurrences would alter the estimated air-sea fluxes would be beneficial.

> **Response: Indeed, what an excellent suggestion! As stated, wind speed plays a pivotal role in determining air-sea gas exchange. Therefore, it is essential to assess the validity of the cited wind speeds. Typically, high wind speeds are consistently observed during the northeast monsoon, in contrast to the relatively lower wind speeds experienced during the summer seasons along the coast of Taiwan. For instance, during the southwest monsoon season, the wind speed is 2.4±0.1 m s$^{-1}$, while it reaches 4.7±0.3 m s$^{-1}$ during the northeast monsoon season, based on over 30 years of data averaged from the southern tip of Taiwan (Center Weather Bureau, Taiwan). The wind speeds utilized in our estimations in this manuscript align closely with previously recorded wind speed patterns. We have incorporated similar assumptions in our revision to bolster the credibility of our estimations.**

> **However, it is worth acknowledging that relying solely on mean wind speeds for estimation might lead to an underestimation of gas exchange rates during episodic high wind speed events. To address this limitation, we would like to emphasize that any conditions affecting wind speed can also impact gas exchange estimations. We hope that this clarification addresses your concerns.**

These are my biggest concerns about the paper, however there were several other issues that I think need to be addressed. I have included more specific comments below.

7. Title – The paper doesn't discuss the impact of coral reef ecosystems on the marine carbon dynamics of Nanwan Bay.

> **Response: Thank you for your valuable comment. We indeed agree with your observation that this manuscript primarily emphasizes marine carbon dynamics within coral reef ecosystems, and not the effects of coral reefs on carbon dynamics (i.e., the "other side of the coin"). As a result, we have made a slight modification to**

**the title, which now reads, "Marine carbon dynamics in a coral reef ecosystem of Southern Taiwan".**

8. Line 51 – I think this statement could benefit from a reference.

   **Response: Thank you for noting this. This statement has been slightly modified to read as follows: "The carbon dioxide (CO2) concentration in marine systems varies in response to both region and season (Fay et al., 2021; Sitch et al., 2015; Schimel et al., 2001)."**

9. Line 57 – I think being more specific about exactly what is meant by hydrological characteristics would be useful.

   **Response: Great suggestion, and we're on board with it. The sentence has undergone a slight modification to read as follows: "The hydrological traits (such as temperature, salinity, upwelling, mixing, etc.) of coastal waters exhibit substantial variation, resulting in variations in surface water $p\mathrm{CO_2}$ even within the same continental shelf."**

10. Line 70 – I would use conversely rather than similarly as the data show the opposite trend to the previously mentioned

    **Response: Thank you. The change has been made in accordance with your suggestion.**

11. Figure 1 – The writing on the inset is too small and too similar in color to the rest of the inset, making it illegible. As the goal of this inset seems to be indicating where Nanwan Bay is with respect to Taiwan, I would recommend using a figure that just showed land and sea, rather than depth, as it's hard to tell where Taiwan is. If the goal is to also show bathymetry over a larger area, I think the inset needs to be bigger. The color bars for both the figure and the inset need labels, and I would encourage the authors to use a more colorblind friendly color map. The 'x' marks denoting the sampling sites should be a different color to the underlying bathymetry.

    **Response: Thank you! We concur that the sampling figure appeared overly intricate. In order to enhance clarity regarding the sampling stations, we have simplified the figure, utilizing solely black and white colors.**

12. Line 85 – I'd consider removing 'may' – carbonate dynamics on coral reefs typically vary substantially.

    **Response: Thank you! We appreciate it, and we have removed the word "may" as suggested.**

13. Lines 90-93 – This sentence is unclear. I think the authors should make it more explicit how physiological changes in resident organisms in response to environmental change can in turn affect seawater carbon levels.

> **Response: Thank you! In this revised version, we've included an example that illustrates how changes in the physiology of resident organisms, triggered by environmental shifts, can subsequently impact seawater $pCO_2$.**

14. Lines 94-108 – This reads as a site description and could potentially be moved to methods.

> **Response: Good point! We concur with your observations. As recommended, this paragraph has been relocated to the "Methods" section.**

15. Line 101 – I would like more detail here regarding what these habitats are and what their relative proportions are.

> **Response: Thank you. The sentence has undergone a slight modification and now reads as follows: "The complex seabed in Nanwan Bay encompasses diverse habitats (such as sandy beaches, rocky shores, & coral reefs) and represents the initial point of interaction with the warm and highly saline Kuroshio Current."**

16. Line 103 – What specific impacts do these upwelling events have on temperature and nutrients?

> **Response: Good point. To address this concern, we have introduced the following sentence: " In the course of the upwelling event, the surface water of Nanwan Bay can drop by >3°C, coupled with a rise in nitrate concentration exceeding 2 μM, as documented by (Chen et al., 2005)."**

17. Lines 111-113 – It's not clear if this sentence is referring to Nanwan Bay or upwelling regions in general. What causes some upwelling regions to be sources and some to be sinks?

> **Response: We regret the ambiguity in the initial statement. The intention of this sentence was to highlight a common phenomenon in coastal upwelling systems. To enhance clarity, we have refined this sentence to read as follows in the revised version: "In the majority of coastal upwelling regions, the ocean absorbs $CO_2$ from the atmosphere (Hales et al., 2005), and this process is intricately linked to increased primary production, which thrives in the nutrient-rich conditions resulting from upwelling."**

18. Line 113 – I would advise the authors to reorder this section here and start by talking about the mechanism by which productivity alters carbon cycling, and then focus on how

an increase in nutrients can enhance these effects. I'd also consider replacing basic productivity with primary productivity throughout the paper, as I believe it's a more widely used term.

> **Response: Thanks! Preceding this sentence, we included an additional statement to clarify that $CO_2$ uptake occurs during primary production. Furthermore, we have consistently employed the term "primary productivity" as advised throughout the text.**

19. Lines 117-121 – As far as I can tell, the authors have not characterized the P/R ratio of the bay but instead use calculated $pCO_2$ values to determine if it's a sink or a flux. As such, I'd suggest deleting this sentence, or changing it to discuss how $pCO_2$ gradients determine if the ocean is a sink or a source. I'm also not sure what is meant by a range of biogeochemical processes – to me this paper does not constrain biogeochemical processes.

> **Response: Thank you for the insightful suggestion. We concur that presenting the P/R ratio as an indicator of whether a region is a $CO_2$ source or sink in the atmosphere is inappropriate. To address this, we have revised the sentences as follows: "The disparity between seawater $pCO_2$ levels ($pCO_2^{seawater}$) and atmospheric $pCO_2$ ($pCO_2^{air}$) serves as a valuable metric for determining whether a marine system functions as a source or sink of carbon. In this context, a positive difference, $pCO_2^{seawater} - pCO_2^{air} > 0$, indicates a carbon source, while a negative difference signifies a carbon sink. Our objective in this study was to ascertain whether Nanwan Bay operates as a net carbon source or sink. To achieve this, we conducted a comprehensive analysis of the marine carbonate system across various spatial and temporal scales." This modification hopefully aligns with your suggestion.**

20. Lines 125-127 – More information needs to be provided about the sampling. How many times was each station sampled each day? What time of day was each sample taken at? Which depths were the samples taken at each station? I think a table reporting this would be greatly beneficial.

> **Response: We greatly appreciate your valuable suggestion. Recognizing the need for enhanced reader comprehension, we have taken steps to furnish more comprehensive details regarding the sampling process. To achieve this, we've introduced a supplementary table (Table S1) that encompasses information such as sampling time, sampling depths, and bottom depth for each individual sampling station, as recommended. We believe that this addition will serve to better elucidate the entirety of the sampling framework employed in this study.**

21. Lines 130 & 127 – How were these accuracies determined? Are these the factory standard

accuracies?

**Response: Indeed, you are correct that these accuracies represent the factory standard. To prevent any potential confusion, we have excluded them in this updated version.**

22. Lines 141-149 – It is not clear to me why pH values are being reported on two different scales.

**Response: Good point, and one raised by the other reviewer, as well. Seawater pH was measured using an automated titration system consisting of a Mettler-Toledo DL53 with a DG-111 electrode. Prior to measurement, the electrode was calibrated using Merck standard buffer solution (NIST) at 25°C. The calibration ranges for pH 4, 7, and 10 were set to fall within the range of 176±30 mV, 0±30 mV, and -176±30 mV, respectively (calibration slope of -56 to -59). Measured pH values were expressed on the NBS scale. This pH value was used for $pCO_2$ calculation in this study.**

**In addition, pH buffers were prepared with a pH value of 8.083 for Tris artificial seawater buffer (2-amino-2-hydroxymethyl-1,3-propanediol) and a pH value of 6.776 for AMP artificial seawater buffer (2-aminopyridine). These two buffer solutions were used to calibrate the electrode for measuring the pH value on the Total scale ($pH_{tot}$). During the calibration process, the slope corresponding to the potential and the pH value should theoretically be above 98% before adoption. After calibration, the electrode was immersed in clean seawater, and the pH value was measured once it stabilized. The measurement process should not take too long to avoid the influence of atmospheric exchange on the sample, which could cause a change in pH value. The pH value of the seawater sample to be tested was measured at 25°C in a constant temperature bath. Despite our confidence in these findings, we have decided to exclude $pH_{tot}$ and rely solely on $pH_{NBS}$ in this revised version to maintain clarity.**

23. Line 157 – Very. Minor, but I believe CO2SYS is spelt without a subscript. I'd also encourage the authors to indicate which version of CO2SYS they have used.

**Response: Thank you for bringing this typo to our attention; it has been rectified at all places in the text. Furthermore, we have now specified that the version of CO2SYS was 1.02 in this revised manuscript.**

24. Line 160 – which measured pH scale was used, NBS or total? Both?

**Response: Thank you for highlighting the ambiguity in our statement. In this study, we used $pH_{NBS}$ for all calculations, and we have made this clarification in this**

**revised version. Additionally, for further details, please refer to our response to comment 22. Briefly, we no longer present the pHtot data.**

25. Line 162 – I believe Dickson and Millero (1987) refit the values from Mehrbach et al. (1973)

> **Response: Indeed, you are correct. To address this, we have made a slight modification to the statement, and it now reads as follows in this revised version: "The dissociation constant of carbonic acid…K1 and K2 values from Dickson and Millero (1987) refit from the values of Mehrback et al. (1973)."**

26. Line 170 – I would keep this section focused on the equation and say where the data came from later on.

> **Response: Thanks. This is a good suggestion. This statement has been subtly adjusted in this revision to maintain its focus on the equation, as suggested.**

27. Line 173 – I would include a citation on this equation.

> **Response: Thanks. As suggested, a citation for this equation has been included.**

28. Lines 177-180 – I believe more justification is needed here as to why samples measured at Dongsha Island are applicable to Nanwan Bay, given that these values directly influence the direction and magnitude of the air-sea $CO_2$ flux. Nanwan Bay's higher proximity to land may mean that anthropogenic and terrestrial effects (i.e., effects from terrestrial growing seasons) alter its $pCO_2$ dynamics in comparison to Dongsha. $pCO_2$ data from somewhere on Taiwan that demonstrated a similar trend to that observed on Dongsha would strengthen this argument.

> **Response: Thank you for your valuable comment. Taiwan currently has only two $pCO_2^{air}$ monitoring stations, namely Dongsha Island and Lulin. In response to your fifth comment, we have included $pCO_2^{air}$ data from the Lulin station for comparison. Our analysis of the data reveals that $pCO_2^{air}$ values at both stations closely correspond, with the exception of some variation evident during the summer months. This divergence in summer values can be attributed to the vigorous growth of terrestrial vegetation near the Lulin station (as expected). Nonetheless, we maintain that this seasonal terrestrial influence is likely to have only a minor impact on Nanwan Bay, especially in light of the persistent southwest monsoon during the summer. This comment also shares similarities with your comments 5, 6, 53, and 54. For further details, please refer to our responses to those specific comments.**

COME BACK TO LINES 189 TO 207

29. Table 1 – It would be good to see timeseries of these data either in the paper or in supplementary materials, rather than just this correlation matrix.

**Response: Thank you for your valuable suggestion. Our intention was to illustrate both the temporal and spatial relationships amongst the carbonate variables and seawater properties in Table 1. Furthermore, we have included time series data for the previous variables at S10 (as opposed to all sampling stations) in Figs. 3, 5, and S1-S5 in this revised version. These figures can be considered as representative time series data for this study.**

**The organization of our results presented here not only maximizes the available information but also streamlines the presentation to save space. We believe this approach is logically sound for our presentation. Nevertheless, if you still feel that presenting all the time series data is necessary, we would be more than happy to accommodate that request.**

30. Line 218-220 – I'm afraid I don't totally understand why these findings indicate vertical mixing in spring and winter and upwelling in spring and would ask the authors to explain this more.

**Response: Thank you for your valuable comment. We acknowledge that the initial statement was too simplistic in addressing why these findings indicate vertical mixing and upwelling. In this revised version, we have made significant improvements by providing more detailed information to thoroughly explain the underlying reasons. For a thorough grasp of the subject, we invite you to refer to the corresponding section in this updated version.**

31. Lines 222-224 – I think the connection between TA and these factors needs to be drawn out more, particularly since riverine outflow following rainfall can have high TA values and so increase TA even as it freshens water. I also think the paper would benefit by discussing specifics related to this site – are there any big rivers that flow into it? What's the seasonal rainfall cycle like?

**Response: Thank you for the valuable suggestion. In this coral reef ecosystem, there is no major rivers nearby, and we did not observe any rainfall events one week prior to each sampling. This indicates that changes in TA may not be linked to freshwater influence during our sampling period. We have addressed these statements in the revision to help clarify your concerns.**

32. Figure 3 – All the font sizes in this figure need to be increased. The color bar should be labeled.

**Response: Thanks. This figure has been revised in accordance with the suggestions.**

33. Lines 231-233 – I'm assuming the means calculated here are spatial, in which case I think care needs to be taken to account for the different distances between sampling points. The closeness of S31 and S33 mean that area will be overweighted if a mean is calculated without any weighting, while the lack of points near S15 would mean that this area will be underweighted. Given that these mean values are what are used to calculate the air-sea flux, I think the authors have to address this, and at the very least need to be clearer about how mean values are calculated. I would also stress that only surface water values were used, and say at what depth surface water values were obtained.

> **Response: We deeply appreciate your thorough examination of our results and your valuable comments. In response to your comment 3, we conducted an evaluation to assess the potential impact of the issue raised. This evaluation considered both temperature and non-temperature effects on surface water $pCO_2$, and it was performed both with and without data from the stations surrounding the nuclear power plant outflow. Based on our findings, it appears that the unevenly spaced sampling regimen does not significantly affect the estimation of mean values, and, therefore, we have decided to include data from both stations (S31 and S33) in this revised analysis. We do acknowledge the importance of clarifying how the mean value is calculated and providing information regarding the depth of the surface water used for this calculation. We will make sure to address this by explicitly detailing the mean value calculation process in the text and the Method section. We hope this clarification will address your concerns. Thank you for your valuable input.**

34. Figure 4 – The font size needs to increase throughout these figures. This figure also implies that each site was sampled multiple times per day – this needs to be mentioned and described in the methods.

> **Response: Thanks! The font size in the figures within this manuscript has been enlarged for better clarity. Additionally, in accordance with the suggestion, the sampling time has been specified in Table S1 and is also mentioned in the legends of the respective figures.**

35. Lines 240-244 – I believe the authors have the reasoning here backwards. Vertical variation is an indication of stratification, or a poorly mixed water column, while well mixed water masses tend to have constant properties throughout the water column.

> **Response: Thank you for highlighting this misleading statement, which the other reviewer also noted as being erroneous. The sentence has been revised to read as follows: "During spring and winter, pronounced mixing was evident, as demonstrated by the straight vertical profiles in temperature and salinity in Fig. 3a, d, e, and h. Conversely, in summer and autumn, mixing was less**

**apparent."**

36. Figures 5-8 – These figures could potentially be collapsed into one by calculating an average profile for each parameter each season, then plotting those average profiles on a single figure.

   **Response: Thanks! We appreciate the feedback, and as suggested, we have consolidated Figures 5-8 into a single figure. This modification has enhanced the clarity of our manuscript, better addressing the study's purpose. The original Figs. 5-8 have been moved to supplement material to become Figs. S1-S4 which serve as supporting evidence.**

37. Figures 5-8 – These figures show pretty high surface $pCO_2$ variability over the course of a day. How is that being accounted for when calculating average $CO_2$ values? Are similar levels of $CO_2$ variability present at other sites?

   **Response: Thanks! As evident from the data, surface $pCO_2$ exhibited significant variability throughout the day, particularly during autumn. Although daily fluctuations were also observed in other seasons, they appeared to be confined to a narrower range. For the sake of comparison, the values at sampling site S10 were averaged over the entire sampling date to represent the entire sampling season. In this study, we exclusively conducted repeated $pCO_2$ sampling over the course of a day at S10. Consequently, we cannot provide insight into the variability of surface $pCO_2$ at other sites on the same day. Nevertheless, it is worth noting that significant diurnal variation in surface $pCO_2$ has been documented in another coral reef ecosystem (Yan et al., 2018).**

38. Lines 272-276, and Figure 10 – I think what $pCO_2$ are at $T_{obs}$ and $T_{mean}$ could be made clearer. As far as I understand, $pCO_2$ at $T_{obs}$ is the $pCO_2$ value you would measure at a given temperature if $pCO_2$ if temperature was the only factor affecting it, while $pCO_2$ at $T_{mean}$ would be the $pCO_2$ you would get by normalizing observations to the annual mean temperature. To me, these alone wouldn't be the temperature and non-temperature effects. Instead, the temperature effects would be $pCO_2$ at $T_{obs}$ – $pCO_2$ at $T_{mean}$, while the non-temperature effects would be the difference between the measured $pCO_2$ and the $pCO_2$ at $T_{obs}$. If this is the case, then I think nT – T just gives you the range between these values – it doesn't tell you about the relative influence of temperature or non-temperature effects. You'd get the same nT – T if the nT value was 5 µatm higher than the mean and the T value was 35 µatm lower than the mean as you would if the nT value was 20 µatm higher than the mean and the T value was 20 µatm lower, but the influence of each effect would not be the same. This would affect a large amount of the analysis that follows (e.g., lines 297-299).

**Response: Thank you for your in-depth examination of our results and discussion. We genuinely value your insightful suggestions, which have spurred us to reconsider our findings. We wholeheartedly concur with your point about the importance of evaluating both temperature and non-temperature effects, as per your recommendation. In this revised version, we have meticulously re-calculated and re-analyzed our data, implementing your suggestions to update Figures 7, 8, and 9. Consequently, we have made corresponding adjustments to the relevant results and discussion sections. This approach has significantly bolstered the strength of our argument concerning these issues.**

39. Line 277 – Is this a spatial mean rather than the mean in the equation above? The annual mean doesn't seem like it should be changing over time. This is covered in the caption to Figure 10, but I think it should be included in the main text.

**Response: Indeed, your observation is accurate. This represents the spatial mean computed at each station throughout the sampling period. We have made slight adjustments to the main text to clarify this.**

40. Lines 281-283 – Why do you believe that? What are possible non-temperature effects? Why would they act in the other direction to temperature?

**Response: In addition to temperature, surface seawater $p$CO$_2$ levels can be influenced by various factors, including gas exchange, tides, currents, river discharge, upwelling, vertical mixing, and biological processes. Some of these factors may contribute to non-temperature effects. However, in our study area, the absence of a large nearby river allows us to exclude river discharge as a significant contributor. Upwelling has been observed in this coral reef ecosystem, which is characterized by a highly productive benthic community. This suggests that upwelling is one of the most likely factors contributing to non-temperature effects on $p$CO$_2$. Furthermore, our analysis revealed a relationship between surface $p$CO$_2$ and phytoplankton biomass (e.g., Chl $a$). This finding suggests that $p$CO$_2$ may also be influenced by phytoplankton metabolism in this particular case. In conclusion, it appears that non-temperature effects on $p$CO$_2$ levels in this study area are likely the result of a combination of factors.**

41. Lines 283-286 – Why would consistently higher water temperatures mean that pCO$_2$ is more variable? Seems like more variable temperatures would lead to greater pCO$_2$ variability.

**Response: That is exactly what we tried to address, and this sentence has been modified slightly as suggested.**

42. Figure 10, and other figures with multiple axes – I think the figures could be made a bit clearer if the axes were colored to match their data. You could change the color of the nT-T and right axis to match each other.

> **Response: Thanks! We appreciate the suggestion, and the figures have been modified accordingly.**

43. Figure 11 – It's not super clear what's meant here by control factors, or where the standard deviations have come from.

> **Response: Thank you for highlighting the ambiguity in the statement. To provide clarity, the figure legend has been revised as follows:**
>
> **Fig. 8. Mean values and impact levels of surface water $pCO_2$ in Nanwan Bay over different seasons. "Mean" represents the average value across sampling stations for each season. "nT" denotes non-temperature effects on surface water $pCO_2$, while "T" signifies temperature effects on surface water $pCO_2$. Vertical lines indicate the standard deviations.**

44. Lines 299-304 – I think there needs to be some discussion of why Chl $a$ influences $pCO_2$ here if the authors are going to claim that Chl $a$ is one of the main factors affecting $pCO_2$ in autumn. Why would an increase in Chl $a$ increase $pCO_2$? It seems plausible that if there was an increase in Chl $a$, then you'd expect a lower $pCO_2$ due to an increase in photosynthesis. Conversely, could it be that an increase in $pCO_2$ increases Chl $a$?

> **Response: Thank you for your valuable input. Your inquiry is thought-provoking, and we wholeheartedly agree on the importance of providing a brief explanation of how Chl $a$ influences $pCO_2$, given the significant relationship observed between Chl $a$ and $pCO_2$. In this revised version, we have incorporated a concise statement addressing the impact of Chl $a$ on $pCO_2$, as per your suggestion. However, it is worth noting that, as mentioned later in our statement, despite the influence of non-temperature factors, particularly Chl $a$, the low $r$ values indicate the likely presence of unmeasured variables contributing to the temporal variation in $pCO_2$. Consequently, we currently lack a reasonable explanation for the observed interplay between increasing Chl $a$ and $pCO_2$ (or vice versa).**

45. Line 300 – This is the first time Chl $a$ has been mentioned. How was it measured? Where was it measured?

> **Response: Thank you for highlighting this oversight. We collected and analyzed Chl $a$ samples from all water samples at each station. The detailed methods for Chl $a$ sampling and measurement have now been included in the respective methods section in this revised version.**

46. Figure 12 a & b – Both of the significant relationships seem to be heavily influenced by single points at higher temperatures and Chl *a* concentration than the remaining points. Are the fits still significant without those points?

> **Response: Wow! Good catch! The statistical significance of the relationships holds within the margin for temperature (albeit marginally; *p*=0.06) and Chl *a* concentration (*p*≤0.05) when the largest data point is excluded. This suggests that the significant relationships should indeed hold in our current estimation.**

47. Lines 312-314 – Doesn't Fig. 12c show $pCO_2$ at $T_{mean}$ rather than at the actual temperature? If it is showing it at actual temperature, then what is 12a showing? I'd also recommend replacing actual temperature with measured temperature, which is how it's been used previously.

> **Response: Thank you for bringing these typos to our attention. In this revised version, they have been rectified, both in the text and within the figure panels.**

48. Lines 313-316 – I don't think Chl *a* itself is an effect, and so I think the authors need to explain the connection further.

> **Response: Thank you for your comment. To connect the relationship between Chl *a* and seawater $pCO_2$, we have included the following sentence in this revision: "In general, Chl *a* influences $pCO_2$ through its involvement in photosynthesis and the subsequent removal of $CO_2$ from seawater by phytoplankton (Chen et al., 2019)." We hope you find this justification reasonable.**

49. Lines 319-323 – It's not clear how this relates to the results of this paper. Is DNC calculated somewhere? How do measured changes in the mentioned parameters compare to what would be expected from DNC? How would these changes alter the chemical and biological processes of the surface water?

> **Response: Thank you for bringing the unrelated statement to our attention. In this revised version, we have removed the mentioned statement per your suggestion.**

50. Line 324 – I think this should be more specific about what exactly the benthic environment of Nanwan Bay is, what is being regenerated, and what processes are leading to it being regenerated.

> **Response: Thank you for your valuable comment. In this statement, we aimed to highlight the general nutrient regeneration processes within the coral reef ecosystem's benthic environment. In this revised version, we have made slight modifications to emphasize these general regenerative processes. We hope this addresses your concerns.**

51. Lines 324-335 – This paragraph seems a little out of place as nutrients have not really been discussed to this point. I think it's a worthwhile addition, but I think it needs some information about how nutrients would influence pCO₂.

**Response: Thank you for supporting our presentation. We agree that providing information on how nutrients affect *p*CO₂ is important. We have made an effort to address how nutrients in this coral reef ecosystem influence phytoplankton and the subsequent uptake of *p*CO₂. To bridge this gap at the beginning of this paragraph, we have added the following sentences: "As mentioned above, seawater *p*CO₂ levels can be influenced by phytoplankton via photosynthesis. Therefore, nutrient availability in seawater primarily affects *p*CO₂ levels by either promoting or limiting phytoplankton growth and consequently primary production."**

52. Line 338 - $\Delta pCO_2 > 0$ doesn't necessarily mean the seawater is supersaturated with $CO_2$, just that $CO_2$ will move from the seawater to the atmosphere.

**Response: Yes, you are correct. This sentence has been slightly modified in this revision to read as follows: 'When $\Delta pCO_2 > 0$, CO₂ in seawater is released into the atmosphere, contributing...'".**

53. Lines 353-356 – I think there needs to be some justification as to why it's reasonable to use the mean values. The gas exchange rate has a quadratic dependence on wind speed, so if there are a few wind events where the maximum wind speed is much greater than the mean wind speed then using the mean could lead to a large underestimation of the gas exchange. As noted in lines 362-363, wind speed is a crucial factor in determining air-sea gas exchange, so I think more justification is required as to why the cited speeds are reasonable.

**Response: Thank you for your valuable comment. This feedback is similar to your comment #6, and we kindly direct you to our response to that specific comment for more information.**

54. I also again think that there needs to be more information about how the mean surface pCO₂ values are calculated – how are they averaged both temporally and spatially? There also appears to be something of a diel cycle in the surface pCO₂ values (e.g., Fig 6d) – how would this influence the variability in the gas exchange rates?

**Response: Thank you for your valuable suggestion. This comment aligns with your previous comments #3 and #4. For further details, please consult our responses to those comments.**

55. Lines 367-373 – What is different about this reef that might make it a sink rather than a source?

**Response: What an insightful comment! It has truly sparked our further contemplating the distinct carbon dynamics of this coral reef ecosystem. Our dataset revealed a $CO_2$ sink during the spring and winter seasons when the water column exhibited robust vertical mixing. In this context, the lower water temperature and subsequently reduced $pCO_2$ levels in the water column became apparent. Moreover, the northeast monsoon season brought higher wind speeds, which further intensified the $CO_2$ sea-air gas exchange flux.**

**In contrast, during the summer, despite the high $\triangle pCO_2$ values, the wind speeds associated with the southwest monsoon were relatively low. Consequently, the $CO_2$ sea-air gas exchange flux into the atmosphere during the summer was insufficient to compensate for the flux sink to seawater observed in the spring and winter.**

**In summary, these findings suggest that the strong vertical mixing and upwelling in spring and winter, resulting in lower seawater temperatures and reduced $pCO_2$ levels, play a pivotal role in transforming this coral reef ecosystem into a carbon sink rather than a source to the atmosphere. This transformation is especially significant during the northeast monsoon season, when reinforcements by strong wind speeds were pronounced. We wholeheartedly agree that this represents one of the most important outcomes of our study, and we have provided a clear explanation of these findings in the revised text.**

56. Lines 373-374 – What are the land-based inputs to Nanwan Bay? What effect might they have?

**Response: Thank you for your comment. When we refer to "land-based inputs," we are primarily discussing the influence of freshwater and its associated components on TA and/or DIC in nearshore environments. However, as we pointed out, there were no significant factors such as a large nearby river or substantial rainfall occurring one week before our sampling. Therefore, the impact associated with land-based inputs is likely to be minor in this specific case.**

**To provide further clarification, we have slightly modified the sentence in question as follows: "The biogeochemistry of nearshore environments may be influenced by land-based inputs, potentially leading to differences in $pCO_2$ and variations in the $CO_2$ sea-air flux. Nevertheless, as previously mentioned, the impacts of land-based inputs are expected to be minimal due to the absence of large rivers in close proximity to this coral reef ecosystem."**

**For more detailed information on this topic, we encourage you to also refer to our response to your comment #31, which addresses a similar issue.**

57. Lines 375-379 – As far as I understand, the wind speed measurements in this paper come from buoys. If this is the case, why is the difference between ship-based and satellite-based estimates of gas exchange relevant to this study?

**Response: Thank you for your comment, and we apologize for the earlier vagueness in our statement. Initially, our intention was to emphasize that, although the trend of the $CO_2$ sea-air flux remained consistent when using both ship-based and satellite-based wind field data for estimation, there were notable disparities in their absolute values. We aimed to convey that our wind dataset was observed from buoys, making it closer to the ground truth situation. Furthermore, we intended to stress the significant impact of wind speed data on $CO_2$ flux calculations.**

**To provide clarity, we have revised the text as follows: "Furthermore, prior studies have shown that ship-based and satellite-based wind field calculations of the $CO_2$ sea-air flux in the East China Sea exhibit similar trends, but significant differences exist in absolute values with ship-based calculations, resulting in greater $CO_2$ exchange compared to satellite-based ones (Tseng et al., 2011). In this study, we utilized wind speed data from buoys situated within the coral reef ecosystem for estimating the $CO_2$ air-sea exchange flux. This approach should offer a more accurate representation of the actual flux *in situ*". We hope this revised statement addresses your inquiry more clearly.**

---

## Author Comment (AC2)

**Response to reviewers' 2 comments on ms no: egusphere-2023-1097 "The impact of coral reef ecosystems and upwelling events on the marine carbon dynamics of Southern Taiwan" (Meng, Chang, Hsieh, Mayfield, and Chen)**

1. This is a study about the carbonate system in a bay of Taiwan. It shows data from four seasons enabling conclusions on the seasonal cycle. Air-sea fluxes of $CO_2$ are estimated and it is concluded that the Nanwan Bay is a $CO_2$ sink, opposite to many other coral reef regions.

   The authors present fluxes of $CO_2$ of the Bay and conclude that the Bay is a sink. However, the uncertainty is large. The uncertainty of the calculated $pCO_2$ was not given and it is likely to be large. Another uncertainty is the atmospheric $pCO_2$, which is from a region far from the Bay.

   **Response: We greatly appreciate your thorough review of our manuscript. We have made substantial revisions in response to your insightful recommendations and those of the other reviewers. Your valuable and constructive comments have not only enhanced the quality of our manuscript but have also inspired us to reconsider how we present and interpret our results.**

   **For your convenience, please find our detailed responses to your comments below, with particular attention to your concerns regarding the uncertainty of the $pCO_2$ calculation. We have addressed these concerns in our responses to your comments numbered 2, 12, and 19, as well as in our responses to Reviewer 1's comments 1-7, which were also centered around this topic. We believe that our revisions adequately address your concerns and have significantly improved the manuscript.**

   **We are now confident that this manuscript meets the standards for publication in *Ocean Science*.**

2. There is a lot of background missing on the calculations, for example, how were the $pCO_2$ values used to produce a number for the area.

   **Response: Thank you for pointing this out. In this revised version, we have made an effort to offer comprehensive background information about our methodology, particularly focusing on the utilization of the $pCO_2$ value to generate or calculate subsequent values. Please refer to our detailed response to your comments numbered 35-37. We trust that our response and revisions adequately address your inquiry.**

3. In the title and in the manuscript upwelling is mentioned. There is no clear method to show or calculate upwelling, which makes all contentions about upwelling vague.

   **Response: Thank you for your valuable comment. We agree with your observation**

**that the discussion of upwelling in the manuscript was not sufficiently detailed. In this revised version, we have expanded the discussion of upwelling and its associated impacts. Additionally, we have slightly adjusted the title to better align with the content of this manuscript.**

**Below is a listing of mostly minor comments:**

4. L59 the coasts of Galicia and Oregon: Why exactly are those mentioned here? Are they representative for other regions?

   **Response: Thank you! We aim to highlight that even within similar systems, such as those experiencing intensive upwelling, their behavior as $CO_2$ sinks or sources can vary. To elaborate further, we have slightly modified the sentence as follows: "Furthermore, upwelling areas along the coast of California and Oman act as $CO_2$ sinks, whereas those along the coasts of Galicia and Oregon serve as $CO_2$ sources (Borges and Frankignoulle, 2002; Friederich et al., 2002; Goyet et al., 1998; Hales et al., 2005)."**

5. L64 delete: can.

   **Response: It has been deleted as suggested**.

6. 72 of the ocean area    (add: area).

   **Response: Thank you; "area" has been added as suggested**.

7. L80 will differ    (not: may differ).

   **Response: Thanks. We agree with you, and "may" has been deleted in this sentence.**

8. L85-88 I think this is a strange sentence. The second part about oceanographic anomalies does not seem to fit with the previous factors.

   **Response: In this revised version, we've incorporated an example that demonstrates how alterations in the physiology of resident organisms, prompted by environmental changes, can subsequently influence seawater $pCO_2$.**

9. L98 delete: but.

   **Response: Thank you once more. We have followed the suggestion and deleted the content. Notably, this paragraph has been relocated to the "2.1 Study Site" subsection within the Methods section.**

10. L109 What is basic productivity? Is it the same as primary productivity?

   **Response: We apologize for any confusion. Yes, we simply meant "primary productivity" here, and have made the corresponding update.**

11. L122, 123 Please use format like 31 March 2011.

**Response: The suggested format has been implemented.**

12. L123 Is the winter cruise in 2013 indeed, i.e., more than one year after the autumn cruise? In that case the data do not show a genuine seasonal cycle; also interannual variability would play a role.

> **Response: You are correct. The winter cruise occurred in 2013. We recognize that the data may not depict a genuine seasonal cycle. Nonetheless, the data collected during the winter cruise can still serve as a representation of the winter period, drawing from our multi-year study of this coral reef ecosystem. For illustration, we provide temperature and salinity profiles at S10 from this study and data from our earlier study on January 5, 2003, for your reference, as shown below. It is evident that the variables may exhibit slight interannual variation, but the patterns remain similar during the winter period in different years.**

[Figure]

13. L125 Is it correct that station S10 is close to the nuclear power outlet? Or are these stations S31 and S33 (also mentioned later in the manuscript).

> **Response: We apologize for any confusion. Stations S31 and S33, rather than station S10, are situated near the Nuclear Power Plant outlet, as stated later in the manuscript. It's possible that you have an earlier version of our submitted manuscript, and this typo has been rectified in our subsequent submission.**

14. 14. L133 and 155 Please also give the precision and/or the accuracy of the oxygen data. In the figures the oxygen concentrations are given in mg/L, which is not usual in oceanography anymore. Usually concentrations are given in (u)mol/kg.

> **Response: The accuracy of dissolved oxygen (DO) has been addressed as suggested. Regarding the DO unit, we currently prefer using mg L$^{-1}$, as it is a unit familiar to most biological oceanographers. However, we are open to changing it to**

**μmol kg⁻¹ if you believe that is more appropriate.**

15. L158 K1 and K2 (capitals).

**Response: Thank you for the reminder. They have now been capitalized.**

16. L166 please use superscript for exponents.

**Response: Thank you for bringing these typos to our attention. They have been corrected in this revision.**

17. L170 please use superscript for exponents.

**Response: We apologize for the typos once more. They have been corrected in this revision.**

18. L171 mol L⁻¹ atm⁻¹ (no moles and use different format).

**Response: Thanks. It has been changed as suggested.**

19. L174 The beginning of the Intro says that the atmospheric $CO_2$ varies significantly based on region. Please justify why you use atmospheric $CO_2$ values that far from the bay.

**Response: Thank you for this comment. We did indeed address the significant variability in atmospheric carbon dioxide concentration based on region and season in the Introduction. Originally, our focus was on understanding the variation in seawater $CO_2$ concentration with respect to both region and season. Consequently, we have slightly modified the sentence to read as follows: "Carbon dioxide ($CO_2$) concentrations in marine systems exhibit notable variation according to both region and season (Fay et al., 2021; Sitch et al., 2015)"**

**You are correct in pointing out that atmospheric $CO_2$ levels can also vary significantly by region, particularly in forested and industrial areas. Although Dongsha Island is a remote location far from our study site, we opted to feature data from there versus those from the Taiwanese mainland (from NOAA) because of the similar latitudes (approximately 21.90°N) and the fact that Dongsha Island is a coral atoll (i.e., dominated by local coral ecosystems). For more details on this matter, please also refer to our response to comment 5 from Reviewer 1.**

20. L178-179 when accessed.

**Response: The data were accessed on 21 November 2022.**

21. L183 … circulation patterns as follows:

**Response: Thanks! The suggested change has been made.**

22. L189 and further: As the explanations are interesting, they are certainly easier to understand when a larger chart with currents etc. would be available. I encourage the authors to provide it.

> **Response: Thank you for the valuable suggestion. We concur that a chart depicting currents would enhance comprehension. However, we want to highlight that the regions mentioned are already included as an inset in Fig. 1, and we believe that this should suffice in allowing readers to follow our description. Therefore, to maintain consistency with our current presentation and to simplify the content of this manuscript, we plan to retain it as is.**

23. L195-200 This is mostly about external influence on the Nanwan Bay. What about local processes?

> **Response: Thank you for noting this. In the introduction, we have already discussed that the most significant local process is the periodic upwelling, which has the potential to influence the hydrographic patterns in Nanwan Bay. Thus, we aim to avoid repeating this statement here.**

24. L203-205 Please explain why this hints at vertical mixing and upwelling.

> **Response: Thank you for your insightful feedback. We recognize that the initial statement may have oversimplified the explanation of why these findings suggest vertical mixing and upwelling. In this updated version, we have enhanced our explanation by offering a more comprehensive and detailed account of the underlying factors. To gain a deeper understanding, we encourage you to consult the corresponding section in this revised document.**

25. L211 Surface $p\mathrm{CO_2}$ levels …

> **Response: Thanks. It has been changed as suggested.**

26. L213 I think a mean value cannot be a range.

> **Response: We apologize for the error. The means (±SD) for these levels have been provided in this revised article.**

27. L214 Are these sea surface temperatures? If yes, that should be mentioned.

> **Response: Yes! This part has been modified slightly to read as "The mean surface seawater temperatures during …"**

28. L219 It is not clear to me why the authors choose especially station S10 as monitoring station. This is exactly the station where the outlet of the nuclear power plant is. Or isn't that the case (see my earlier comment about this at L125).

> **Response: Thank you for your comment. We selected station S10 as the monitoring**

**station due to its significant depth, 105 meters, making it the deepest among all the sampling stations. Our initial intent was to leverage the hydrographic dynamics at this deep station to assess the origins of water masses, whether they originated from the South China Sea, the Kuroshio, or a mixture of the two, as well as to study factors like upwelling intensity and the carbonate parameters of deeper waters. We have elaborated on this intent within the manuscript itself and hope that this explanation addresses your query.**

29. L218 What do you mean with gradient changes? Which gradients?

    **Response: Thank you for highlighting the ambiguity in our statement. We were referring to the changes in temperature and salinity gradients, as evidenced by the depth profiles at Station S10. To provide a clearer explanation, we have made a slight modification to the sentence, which now reads as follows: "Due to the mixing of different water masses by monsoons, tides, eddies, upwelling, and other ocean currents, significant variations in temperature and salinity of the water column were observed at different times at station S10 (Fig. 3) and in the carbonate parameter data (Fig. 5d, S1-S4).". This revision aims to improve clarity.**

30. L220-222 The authors state that in summer and autumn more mixing occurs because of vertical variations in the profiles, while in spring and winter this is less, because the profiles are straight. My interpretation would be the opposite. When the profiles are straight, there has been much mixing. When the profiles are not, mixing is not strong enough.

    **Response: Thank you for highlighting this potentially misleading statement. The sentence has been revised to read as follows: "During spring and winter, pronounced mixing was evident, as demonstrated by the straight vertical profiles in temperature and salinity (Fig. 3a, d, e, and h). Conversely, in summer and autumn, mixing was less apparent."**

31. L224-226 Why is Figure 9 done with station S1 and not with S10, like all previous data? It would be important to show and mention whether such figures can also be made for the other stations, or whether this station is special.

    **Response: Thank you for your valuable comment. Periodic upwelling events have been observed in Nanwan Bay. In the accompanying figure, we have presented data illustrating the relationships between various variables, including temperature, salinity, dissolved oxygen (DO), and pH, throughout a complete upwelling cycle in Nanwan Bay. These data were obtained from a real-time water quality monitoring system at station S1 in our previous study.**

**It is important to note that in our study, water sampling at each station was conducted discretely and was not synchronized within a complete upwelling cycle during the sampling periods. Therefore, the relationships observed in this figure may not be replicated at each station in our study due to the differences in sampling times and locations. Nonetheless, we have included this figure as an example to illustrate the hydrography and its patterns during an upwelling event.**

32. L227 Why are low values evidence for upwelling? Which low values do you mean? Which $pCO_2$ increases do you mean. Please explain better.

> **Response: Thank you for your comment. In response to your previous feedback, we plan to leverage the relationships between various variables observed during the entire upwelling event at S1 from our prior study to help explain the potential presence of upwelling-related bottom water observed in this study. The shared characteristics of the bottom water resulting from upwelling were similarly identified during the spring of this study, and we have clearly presented the associated values in this revised article. To provide further clarification, the sentence has been amended as follows: "Seawater quality profiles of S10 provide additional supporting evidence of upwelling, as indicated by the presence of low temperatures (23.3±0.6℃), low pH (8.16±0.01), high salinity (34.32±0.03), and relatively low $pCO_2$ (385.4±5.4 µatm) across the well-mixed water column in the spring (Figs. 3 and 5)". We hope this addresses your inquiry.**

33. L278-279 Please provide a reference for this contention.

> **Response: Thanks for the reminder. A reference has been cited as suggested.**

34. L280-281 ditto.

> **Response: Thanks again. A reference has been inserted as suggested.**

35. L304-305 More info is needed how the del-$pCO_2$ for the Bay was calculated. Is it just a mean of all stations?

> **Response: Thank you for your comment. We calculated the $\Delta pCO_2$ for Nanwan Bay by averaging data from all 17 sampling stations, despite some uneven distribution. Remarkably, excluding certain stations from the analysis did not significantly affect the overall mean value for the bay. Thus, we are confident that these sampling stations are sufficient for providing an accurate representation of the bay's status. For more details on this matter, please also review our responses to your comment 37 and Reviewer 1's comment 3.**

36. L314-315 What is the exact unit of the absorption? Tonne C or tonne $CO_2$? It may be better to use exponents of 10.

> **Response: Thank you for highlighting this previously unclear statement. The unit in question is tC, and we have made sure to reflect this clearly in the revision. Regarding the value expression, we prefer to maintain its current format as it is more convenient for comparing values across different seasons. However, we are open to using exponents of 10 for the values if you believe that would be a better way to express them.**

37. L314-315 Again, how exactly was the result obtained?

> **Response: Thank you for addressing this important methodological concern. In this revision, we have included a description of how we estimated the bay's area and calculated the seasonal $CO_2$ fluxes in the Methods section. For your reference, this information is presented as follows: "The bay stretches between Cape Moubitou and Cape Oluanpi, covering an approximate area of 30 km$^2$, as estimated using Google Earth Pro." Additionally, "The seasonal fluxes over the Bay were calculated by multiplying the mean $CO_2$ exchange flux at all stations for each season by the bay's area of ~30 km$^2$." We trust that this clarification addresses your inquiry. For more details on this matter, please also review our responses to Reviewer 1's comments 53 and 54.**

38. Figure 1 The inset is not very clear. Also, it is not explained in the caption what all the lines and different arrows mean. I think a simpler map would be better.

> **Response: We appreciate your comment. To enhance the clarity of the sampling station, we have simplified this figure, and all symbols have now been clearly explained in the caption.**